# STYLEALIGN: ANALYSIS AND APPLICATIONS OF ALIGNED STYLEGAN MODELS

**Zongze Wu**
The Hebrew University

**Yotam Nitzan**
Tel-Aviv University

**Eli Shechtman**
Adobe Research

**Dani Lischinski**
The Hebrew University

## ABSTRACT

In this paper, we perform an in-depth study of the properties and applications of *aligned generative models*. We refer to two models as aligned if they share the same architecture, and one of them (the *child*) is obtained from the other (the *parent*) via fine-tuning to another domain, a common practice in transfer learning. Several works already utilize some basic properties of aligned StyleGAN models to perform image-to-image translation. Here, we perform the first detailed exploration of model alignment, also focusing on StyleGAN. First, we empirically analyze aligned models and provide answers to important questions regarding their nature. In particular, we find that the child model's latent spaces are semantically aligned with those of the parent, inheriting incredibly rich semantics, even for distant data domains such as human faces and churches. Second, equipped with this better understanding, we leverage aligned models to solve a diverse set of tasks. In addition to image translation, we demonstrate fully automatic cross-domain image morphing. We further show that zero-shot vision tasks may be performed in the child domain, while relying exclusively on supervision in the parent domain. We demonstrate qualitatively and quantitatively that our approach yields state-of-the-art results, while requiring only simple fine-tuning and inversion.

## 1 INTRODUCTION

Transfer Learning (TL) refers to the process in which a *parent* model, pretrained for some source domain/task, is used to improve the performance of a *child* model on a different target domain and/or task (Pan & Yang, 2009). The assumption underlying TL is that some knowledge learnt by the parent model is transferable to the new domain or task (Pan & Yang, 2009; Torrey & Shavlik, 2010; Yosinski et al., 2014). The most common TL approach is fine-tuning, where the parent's parameters are used to initialize those of the child. Next, the child's parameters, or sometimes just a subset of them, are trained on the target domain/task. Once TL is completed, the child posseses some of the parent's knowledge, despite the fact that the model parameters may have changed.

Existing TL literature typically examines the performance of the child model, e.g., in terms of classification accuracy (He et al., 2019), or FID score (Karras et al., 2020a), without paying much attention to the relationship between parent and child models, induced by the transfer process. Typically, the child model is simply applied to the task it was trained on, while the parent model is no longer used, having fulfilled its purpose. In this work, we provide a complementary perspective, which focuses on analyzing and leveraging the shared knowledge between the two models. Specifically, we consider the case where the TL is performed by fine-tuning the same architecture. We refer to models obtained in this manner as *aligned models*.

Several recent works (Pinkney & Adler, 2020; bryandlee, 2020; Kwong et al., 2021; Song et al., 2021; Gal et al., 2021) use aligned models in a novel manner. In all cases, an unconditional GAN, specifically StyleGAN2 (Karras et al., 2020b) is fine-tuned from domain $A$ to domain $B$. However, instead of applying the child model as an unconditional generator, it is used in conjunction with the parent model to form an image translation pipeline. First, an image from domain $A$ is embedded into the latent space of the parent StyleGAN2 model. The resulting latent code is then fed either into the child model (bryandlee, 2020; Song et al., 2021; Gal et al., 2021), or into a hybrid model created by *layer swapping*, i.e., by combining layers from the parent and the child (Pinkney & Adler, 2020; Kwong et al., 2021).

These methods have achieved great results in image-to-image translation between several domains. Most notably, translating real human face images to a variety of styles such as cartoons and oil paintings (Pinkney & Adler, 2020; Kwong et al., 2021; Song et al., 2021), but also translating humans to dogs and cats to wildlife (bryandlee, 2020; Gal et al., 2021). However, they focus on a specific application (image-to-image translation) and do not explore or leverage aligned models further. As a result, many questions arise but remain unanswered. For example, which parts of the network change in the TL process, and which knowledge is inherited by the child from its parent? To which degree do the answers to these questions depend on the similarity between the parent and child domains? And, is knowledge not used by the child model completely lost or could it be recovered? Finally, what further applications, besides image-to-image translation, can be solved using aligned models?

In this work, we delve deeper into model alignment. In light of previous works, we specifically focus on the state-of-the-art unconditional GAN architecture, StyleGAN2 (Karras et al., 2020b). The process of obtaining aligned models is incredibly simple: we start with a parent StyleGAN2 model trained on domain $A$ and fine-tune it fully for domain $B$, yielding an aligned child model.

We divide the investigation of model alignment into two parts. First, in Section 3, we perform the first empirical analysis of the phenomenon, answering the questions posed above, as well as others. This analysis provides some surprising and novel insights that shed light on aligned StyleGAN2 models. For example, we discover that when fine tuning to a similar target domain, the parts of the model that change the most are the feature convolution weights in the synthesis network. In contrast, the changes in the mapping network and affine layers are negligible (see Figure 1). This crucially implies that the learned latent spaces $\mathcal{W}$ and $\mathcal{S}$ are barely affected by the fine-tuning. This explains our next discovery – that semantically meaningful directions in the latent space of the parent model, retain the same (or similar) semantics in the child model (see Figure 2). As the data domains become more distant, the mapping network and affine layers become more affected, which results in a weaker degree of *semantic alignment*. However, even in extreme cases, such as human faces and churches, some semantic alignment occurs. Another surprising discovery is that the semantic latent controls that seemingly disappear after transfer to the child model, are in fact merely hidden, rather than forgotten, and reappear if the child is retrained back to the parent's domain.

Second, in Section 4, we use aligned models to solve several popular Computer Vision and Computer Graphics tasks. We start with the aforementioned image-to-image translation task (Section 4.1), examine a number of alternatives, and show that aligned models obtain state-of-the-art results for a variety of scenarios. This is especially impactful, as using aligned models for image translation is incredibly simple compared to dedicated methods for the same task, each devising its custom architecture and losses. Next, we explore additional tasks, for which aligned models have not been used before. In Section 4.2 we describe a simple method for fully automatic image morphing between fairly dissimilar domains, such as human to dog faces, which previously necessitated sophisticated methods (Aberman et al., 2018; Fish et al., 2020). Examples of smooth morphs are included in the accompanying video. In Section 4.3, we use aligned models to solve zero-shot classification and regression tasks in domain $B$, where the supervision is available strictly in domain $A$. Conceptually, our method reduces a task in a zero-shot or few-shot setting to the same task in a different data domain where supervision is plentiful.

In summary, while several previous works took advantage of aligned models implicitly, ours is the first work to conduct a thorough empirical study of this phenomenon. Our study reveals various interesting properties that we then use to further leverage aligned models for a variety of applications, almost effortlessly achieving state-of-the-art performance.

## 2 RELATED WORK

**Latent Space of GANs:** With the rapid evolution of GANs (Goodfellow et al., 2014) in recent years, understanding and controlling their latent representation has attracted considerable attention. Specifically, it has been shown that the intermediate latent space of StyleGAN (Karras et al., 2019; 2020b;a) possesses appealing properties, such as being semantically rich, disentangled and smooth. Many recent works have proposed methods to interpret the semantics encoded in that space and its extensions and apply them to image editing (Jahanian et al., 2019; Shen et al., 2020a; Härkönen et al., 2020; Tewari et al., 2020; Abdal et al., 2020; Wu et al., 2020; Patashnik et al., 2021).

In order to benefit from these properties in real images, it is necessary to obtain the latent code from which a pretrained GAN can reconstruct the original input image. This task, commonly referred to as *GAN Inversion*, has been tackled by numerous recent works, either by using: (i) optimization (Abdal et al., 2019; Karras et al., 2020b); or (ii) an encoder (Guan et al., 2020; Pidhorskyi et al., 2020; Richardson et al., 2021; Tov et al., 2021); or (iii) a hybrid approach using both (Zhu et al., 2016; Baylies, 2019; Zhu et al., 2020a). See Xia et al. (2021) for a more thorough review.

**Image-to-Image translation:** The seminal pix2pix work by Isola et al. (2017), first introduced the use of conditional GANs to solve various supervised image-to-image translation tasks. Since then, their work has been extended to allow image synthesis in various different settings: high-resolution (Wang et al., 2018a), semantic image (Park et al., 2019; Zhu et al., 2020b; Liu et al., 2019b), multi-domain (Choi et al., 2018), multimodal (Zhu et al., 2017b), and using a pre-trained generator (Nitzan et al., 2020; Richardson et al., 2021; Luo et al., 2020). Another scenario that has received significant attention is unsupervised image-to-image translation (Liu et al., 2017; Zhu et al., 2017a; Kim et al., 2017; Choi et al., 2020; Lee et al., 2020b), where no paired data samples are given.

Regardless of the setting, all of the aforementioned works train an neural network, designed explicitly for the translation task. Recently, several works (Pinkney & Adler, 2020; bryandlee, 2020; Kwong et al., 2021; Song et al., 2021; Gal et al., 2021) have taken a different approach towards image-to-image translation. They observe that significant correspondence between generated images in different domains exists when an unconditional generator, such as StyleGAN2 (Karras et al., 2020b), is fine-tuned between the two domains. Accordingly, these works take a two-step approach towards image-to-image translation. First, they invert a given image into the latent space of Style-GAN in domain $A$ and then forward the output latent code through a StyleGAN model for domain $B$. The latter model is obtained either by directly fine-tuning from the former model (bryandlee, 2020; Song et al., 2021; Gal et al., 2021) or by *layer swapping* (Pinkney & Adler, 2020; Kwong et al., 2021), i.e., forming a model whose layers are partially those of the fine-tuned model and partially those of the model for domain $A$.

In this work, we delve deeper into this phenomenon, which we refer to as *model alignment*, and go beyond the image-to-image translation task. For example, we demonstrate that the alignment property goes beyond high-level properties, such as pose, and that multiple fine-grained latent semantics are also aligned. We leverage this property for tasks such as morphing and zero-shot classification.

**Fine-tuning and Catastrophic Forgetting:** Fine-tuning was proven advantageous across fields, settings and tasks and therefore became a standard practice in the deep learning literature. Prominent advantages of fine-tuning are enabling few-shot tasks such as classification (Chen et al., 2019) and unconditional generation (Wang et al., 2018b; Mo et al., 2020; Wang et al., 2020; Li et al., 2020; Ojha et al., 2021), improved performance in a wide variety of tasks (Devlin et al., 2018; Radford et al., 2018; He et al., 2020) and faster training convergence (Wang et al., 2018b; He et al., 2019).

While fine-tuning can be an effective technique for solving a new task, it has been well known for over 30 years (McCloskey & Cohen, 1989) that in the process the model "forgets" how to solve the original task, a phenomenon referred to as Catastrophic Forgetting (CF). For example, once a GAN for a certain domain $A$, is fine-tuned to another domain $B$, the resulting model can only generate images in domain $B$ (Seff et al., 2017; Zhai et al., 2019). In settings such as continual learning and multi-task learning, CF is undesirable. In recent years, there has been progress in mitigating it using dedicated methods (Kirkpatrick et al., 2017; Kemker et al., 2018). CF has been also studied in the context of GANs (Liang et al., 2018; Li et al., 2020; Thanh-Tung & Tran, 2020).

Aforementioned previous works devised methods to obtain a better child model using fine-tuning. From a fine-tuning perspective, this means the model would perform better on the new task. From a CF perspective, this means the model's performance on the previous task should not be impaired. We differ from these works significantly, as we make no deliberate effort to affect what happens during training of the child model. Instead, we investigate the relationship between the parent and child models after naïve fine-tuning, and then use it to solve a variety of applications.

## 3   ANALYSIS OF ALIGNED STYLEGAN MODELS

As explained earlier, several previous works observed that a significant correspondence exists between images in different domains, generated from the same latent code by a parent StyleGAN2

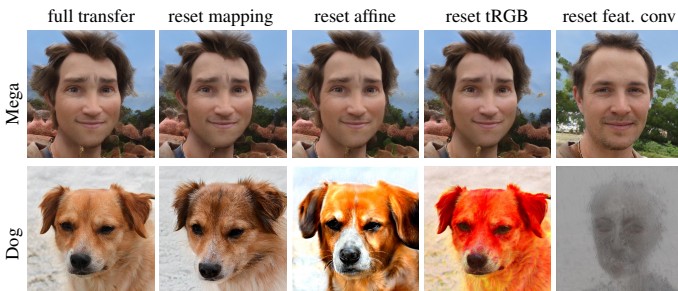

Figure 1: Effect of resetting the weights of different components in child models (Mega, Dog) to their initial values, which come from the parent model (FFHQ). Resetting the feature convolution weights causes the most drastic changes. Also see Figure 8.

model and a child model obtained from it via fine tuning. Our goal is to further understand the relation between the parent and the child models. Below, we explore several aspects.

**Which parts of the network change during transfer?** Recall that the StyleGAN2 model is composed of a mapping function ($\mathcal{Z}$ to $\mathcal{W}$), affine transformations ($\mathcal{W}$ to $\mathcal{S}$), feature convolution layers, and tRGB convolution layers that transform feature maps to RGB images. We transfer a parent StyleGAN2 model pretrained on FFHQ to the Mega cartoon dataset (Pinkney & Adler, 2020) and to AFHQ dog faces dataset (Choi et al., 2020), using ADA (Karras et al., 2020a). After the transfer, we reset the weights of each of the above components in the child models (Mega, Dog) to their initial values in the parent model. The results of this experiment are shown in Figure 1. We observe that the greatest effect on the generated results is caused by resetting the feature convolution layers, which changes the content and structure. Resetting the weights of other components, results in milder changes in both children. This implies that feature convolution layers change the most during transfer. The results also suggest that for the dog model, the affine and tRGB layers have changed significantly more than for the cartoon model. We attribute this difference to the distance between the data domains, and additional experiments in the appendix (Figure 8) support this hypothesis.

While resetting the mapping network has a stronger effect on the dog model, note that the changes are fairly subtle in both datasets, implying that the mapping network changes very little. Effectively, this means that the same $z \in \mathcal{Z}$ is mapped to similar codes in the $\mathcal{W}$ spaces of the parent and the child; in other words, the two $\mathcal{W}$ spaces are *point-wise aligned*. This is a crucial observation as it explains the success of previous works (Pinkney & Adler, 2020; bryandlee, 2020; Kwong et al., 2021) in performing image translation based on aligned models. Simply put, the two latent spaces may be viewed as a single shared latent space. Thus, inversion serves as an encoder from the source domain to this latent space, and the generator is a decoder to the target domain. Viewed in this light, alignment-based image translation resembles several previous image translation approaches (Liu et al., 2017; Huang et al., 2018; Liu et al., 2019a), which are based on shared latent spaces.

**Semantic alignment for similar domains.** In addition to point-wise alignment, we find that the $\mathcal{W}$ and $\mathcal{S}$ latent spaces of the child model are also *semantically aligned* with those of the parent model. By semantic alignment, we refer to the property that latent space controls that affect various semantic attributes of images generated by the parent, have the same (or analogous) effect in the child model. This phenomenon is demonstrated below, both qualitatively and quantitatively.

We demonstrate alignment on closely related domains, by first fine-tuning a parent pretrained on FFHQ (Karras et al., 2019) to the Mega cartoon face dataset (Pinkney & Adler, 2020) and the Metface portrait dataset (Karras et al., 2020a). Next, we apply a variety of latent semantic controls learnt by the parent to the child models. The controls are either individual channels in StyleSpace $\mathcal{S}$, identified by Wu et al. (2020), or directions in $\mathcal{W}$ space, from InterFaceGAN (Shen et al., 2020b). We manipulate images using these controls "as is" in the parent and child models. The initial latent code is obtained by inverting a real image with an e4e encoder (Tov et al., 2021). As may be seen in Figures 2 and 9, regardless of the edited property, or the latent space used, the semantic controls affect the parent and the child models in exactly the same manner. Also see Figures 10 and 11.

To perform a quantitative evaluation we measure the alignment by calculating the overlap between semantic controls found independently in the parent and child models. Since latent directions in $\mathcal{W}$ are affinely related to channels in $\mathcal{S}$, we only examine overlap between style channels. Concretely, we follow Wu et al. (2020) to discover localized channels in both models, and report the number of localized channels for each semantic region in Table 1(a). As can be seen, there is consistently large

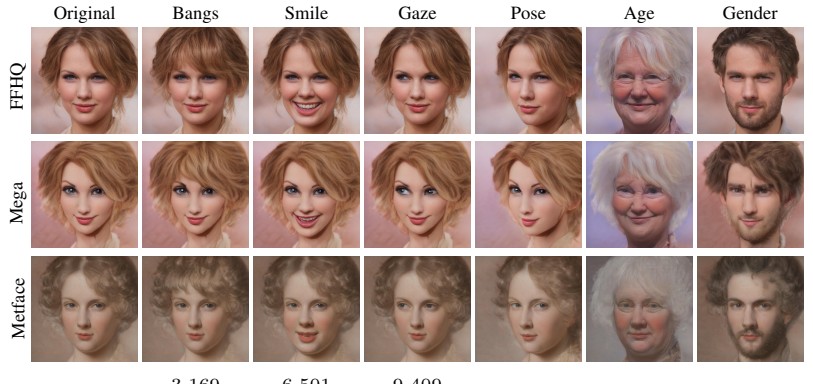

Figure 2: Semantic controls discovered for a parent FFHQ model retain their function in the children models (Mega and Metface). This holds for individual channels in $\mathcal{S}$ (bangs, smile, gaze), as well as for directions in $\mathcal{W}$ (pose, age, gender).

(a) FFHQ2MetFace

|  |  | eyebrow 19 | eye 5 | ear 41 | nose 21 | mouth 32 | neck 46 | cloth 34 | hair 62 |
|---|---|---|---|---|---|---|---|---|---|
| eyebrow | 35 | 15 |  |  |  |  |  |  |  |
| eye | 51 |  | 4 |  |  |  |  |  |  |
| ear | 19 |  |  | 16 |  |  |  |  |  |
| nose | 39 |  |  |  | 14 |  |  |  |  |
| mouth | 14 |  |  |  |  | 6 |  |  |  |
| neck | 24 |  |  |  |  |  | 18 |  |  |
| cloth | 27 |  |  |  |  |  |  | 13 |  |
| hair | 10 |  |  |  |  |  |  |  | 10 |

(b) FFHQ2Dog

|  |  | eyebrow 19 | eye 5 | ear 41 | nose 21 | mouth 32 | neck 46 | cloth 34 | hair 62 |
|---|---|---|---|---|---|---|---|---|---|
| torso | 25 |  |  |  |  |  |  | 4 |  |
| eye | 3 |  |  |  |  |  |  |  |  |
| ear | 142 |  |  | 2 |  |  |  |  | 13 |
| nose | 81 |  |  |  | 4 | 10 |  |  |  |

Table 1: Number of localized StyleSpace channels for various semantic regions. Each column corresponds to a semantic region in parent model, and each row to a semantic region in child model. The number of localized channels that are shared between parent and child are in the center (an empty space denotes 0). (a) After transferring from natural face to portrait, a number of localized channels retain their functions in the same areas (large values on the diagonal), rather than changing their function to other areas (all zeros except diagonal). (b) Even when transferring between more distant domains (human to dog face), we can see that multiple channels retain their function in the same areas (nose, ear), or shift to semantically corresponding areas (from human clothes to a dog's torso, from human hair to dog's ears). Note that the dog face segmentation have no mouth region.

amount of overlapping channels in the same semantic region. We verify that this overlap is not coincidental: performing the same experiment for two unaligned FFHQ models (trained from different random initializations) shows that they have much fewer overlapping channels (see Table 4).

To the best of our knowledge, we are the first to quantitatively measure the fine-grained semantic alignment phenomenon. Our experiments indicate that aligned models for related domains are indeed strongly *semantically aligned*. This phenomenon enables many applications based on transferring knowledge and supervision between aligned models. E.g., zero-shot editing as demonstrated in Figure 2 and zero-shot classification/regression as discussed in Section 4.3.

**Semantic alignment for more distant domains.** To examine the degree of semantic alignment across a wider domain gap, we consider StyleGAN2 models transferred from FFHQ to AFHQ dog faces (Choi et al., 2020). Figure 3 demonstrates that, even in this case, there are still multiple single-channel controls that retain their semantic meaning (e.g., big eyes, black hair, short hair). Furthermore, there are also multi-channel editing directions in latent space that exhibit the same behavior, such as curly hair or small face (from StyleCLIP (Patashnik et al., 2021)), as well as pose (from InterFaceGAN (Shen et al., 2020b)). This appears to be the case for visual attributes that are common to both domains, while controls for attributes that are not present in the target domain (such as glasses, lipstick, or beard) seem to have no effect on the child model. However, as we discuss later, the relevant knowledge is not lost; rather, it is only hidden.

The retained controls reflect some interesting analogies between the domains: for example, controls for hair color and curliness in humans, control fur color and curliness in dogs, while hair length translates to length of dog ears. Interestingly, psychologists have also observed a correlation between

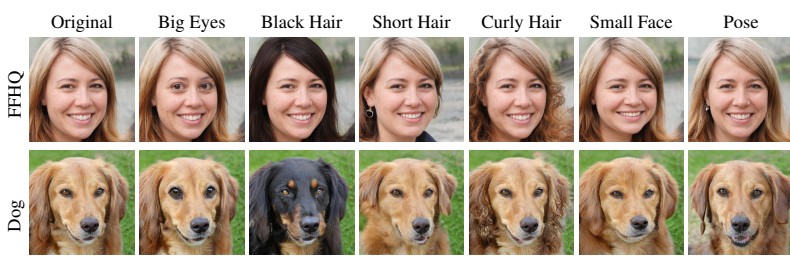

Figure 3: Semantic alignment between single-channel and multi-channel controls for more distant domains (humans and dogs). See also Figure 12 and supp. videos.

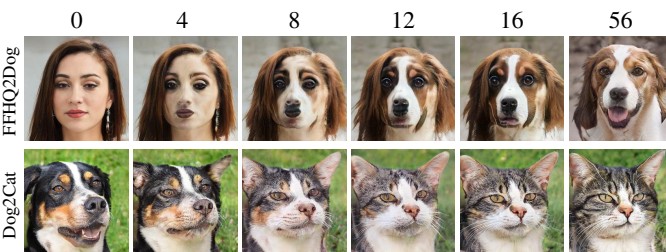

Figure 4: A smooth transition in images generated from the same latent code $z \in \mathcal{Z}$ during fine-tuning. The epoch number appears above each column. The most significant visual changes occur in early epochs (0–16). Also see Figure 13 and supplementary videos.

the hair length in women and the ear shape of their preferred dog breeds (Coren, 1999), which is consistent with the folk belief that people look like their dogs. The gradual emergence of some of these analogies is clearly revealed when examining the samples generated by the model as it evolves during the transfer process. Figure 4 shows images obtained for the same latent vector $z \in \mathcal{Z}$ (in each row), as the training progresses. In the top row, we can see how human hair gradually evolves into dog ears, and the human nose and mouth gradually evolve into the dog's nose and muzzle, while the pose remains mostly unchanged. A similarly smooth transition may be observed when transferring from AFHQ dogs to cats, as shown in the bottom row.

Using the same quantitative evaluation method as before, we further quantify the alignment between a parent FFHQ model and a child AFHQ dogs model. Results are displayed in Table 1(b). As can be seen, a smaller number of channels preserve their semantics when transferred to AFHQ dog as compared to MetFace. Nevertheless, we still observe semantic alignment, albeit weaker, as human ears and hair overlap with dog ears, human cloth control the dog torso, etc.

We next experiment with even farther domains, with barely any similarity between parent and child, such as human faces and churches, which were also examined by Ojha et al. (2021). Despite lack of commonality, the latent direction that controls face pose in the parent still controls the church pose in the child model (see Figure 14). We further examine a double transfer, with FFHQ as parent, AFHQ dog as child and LSUN bedroom as grandchild. The pose direction in FFHQ still controls the pose in the grandchild bedroom model, as shown in Figure 14.

**Are latent semantics forgotten or hidden?** As shown in Table 1(b), when transferring between distant domains, only a small portion of the localized controls retain a similar semantic function. An interesting question that arises is: are the remaining controls completely "forgotten" during the transfer learning, or do they simply become inactive? To examine this, we retrain the child AFHQ dog model back to the FFHQ domain, thereby obtaining a grandchild model, and report the alignment between the original parent and the grandchild models (both for FFHQ) in Table 3. It may be seen that the effect of many of the localized controls are restored. For example, out of the 41 channels that control the ears in FFHQ, only 2 retain a similar function in AFHQ dogs, but 20 regain their function in the grandchild model. This implies that these channels were merely hidden, but not forgotten, during the first transfer learning stage. It should be emphasized that there is barely any such alignment between two *unrelated* models, even when they are trained on the same dataset, as shown in Table 4. Thus, the significant alignment between parent and grandchild (in Table 3) cannot be attributed to re-learning when fine tuning from the child to the grandchild.

**Locality bias in semantics transfer.** We explore this aspect and conclude that only some of the semantic alignment can be attributed to locality bias (see the discussion in appendix Section A.3).

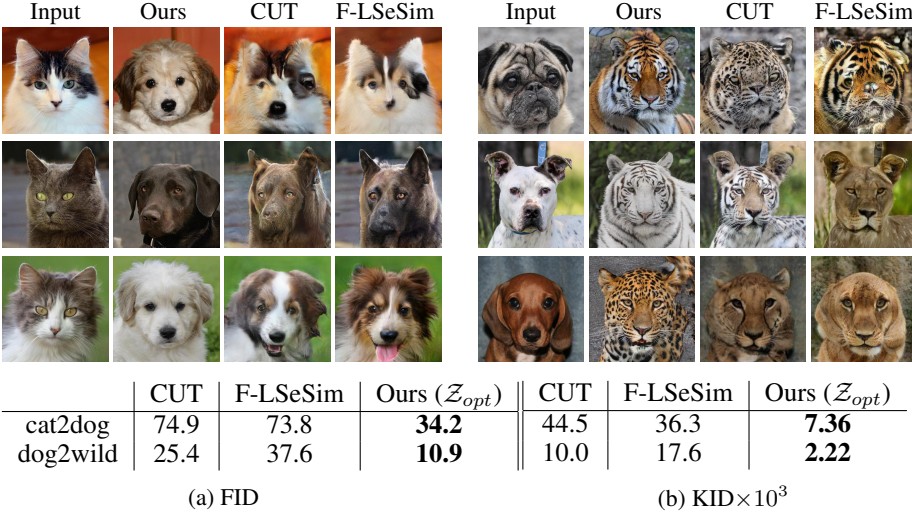

| | CUT | F-LSeSim | Ours ($\mathcal{Z}_{opt}$) | CUT | F-LSeSim | Ours ($\mathcal{Z}_{opt}$) |
|---|---|---|---|---|---|---|
| cat2dog | 74.9 | 73.8 | **34.2** | 44.5 | 36.3 | **7.36** |
| dog2wild | 25.4 | 37.6 | **10.9** | 10.0 | 17.6 | **2.22** |

(a) FID            (b) KID$\times 10^3$

Figure 5: Comparison of I2I translation (cat2dog and dog2wild in the AFHQ dataset) with two state-of-the-art methods. Our method generates realistic target domain images that capture the pose from the source image. In contrast, both CUT and F-LSeSim fail to generate realistic images since they follow the shape of the source domain image too closely. A quantitative comparison in the table below indicates our method is superior by a wide margin, in both FID and KID.

## 4 APPLICATIONS

We next apply aligned models to solve three kinds of tasks: image-to-image translation (Sec. 4.1), cross-domain image morphing (Sec. 4.2) and zero-shot classification and regression (Section 4.3). Efficient training of generators for different resolutions is described in the appendix (Sec. A.4).

### 4.1 CROSS-DOMAIN IMAGE TRANSLATION

As demonstrated earlier, aligned models generate images with similar high-level semantic attributes, given the same latent code. This makes it trivial to translate images between the domains of the parent and the child models, even when these domains are more distant than realistic faces and cartoons or paintings of human faces. For example, it is easy to translate between faces of different species, which typically involves significant changes in both structure and appearance. Furthermore, there's no need for task-specific training or losses; all that is needed is a pair of aligned models and an inversion method to embed real images into the latent space of the source domain StyleGAN.

In Section A.5 we perform a systematic study of which inversion methods (encoder or latent optimization), and which latent spaces ($\mathcal{W}/\mathcal{W}+/\mathcal{Z}/\mathcal{Z}+$), are most effective for image translation. Some previous works (Pinkney & Adler, 2020; Kwong et al., 2021) that considered only similar domains have used the $\mathcal{W}/\mathcal{W}+$ spaces. We find that $\mathcal{Z}$ space yields same level of results for similar domains, but superior results for distant domains, qualitatively and quantitatively. This could be directly explained with a previous observation. For both settings the $\mathcal{Z}$ space is trivially shared, as it is a non-learned space. However, only for similar domains are the $\mathcal{W}/\mathcal{W}+$ spaces aligned and shared.

In Figure 5 we compare our I2I results to two state-of-the-art methods, CUT (Park et al., 2020) and F-LSeSim (Zheng et al., 2021). It may be seen that our method produces realistic and natural looking results, while these two previous methods exhibit severe artifacts, and attempt to follow the shape in the source image too closely, yielding unrealistic results. The table in Figure 5 provides quantitative support for our qualitative observations, yielding significantly lower FID and KID scores for both cat2dog and dog2wild translations. Figure 21 demonstrates our method's ability to perform image translation between dissimilar domains.

In addition to the I2I scenario examined above, aligned models are also able to perform *reference-based* image translation, where the resulting image combines the content of a source image with the style from a second (reference) image (Huang et al., 2018; Choi et al., 2020). StyleGAN inherently supports content and style disentanglement through style mixing. Specifically, we combine the

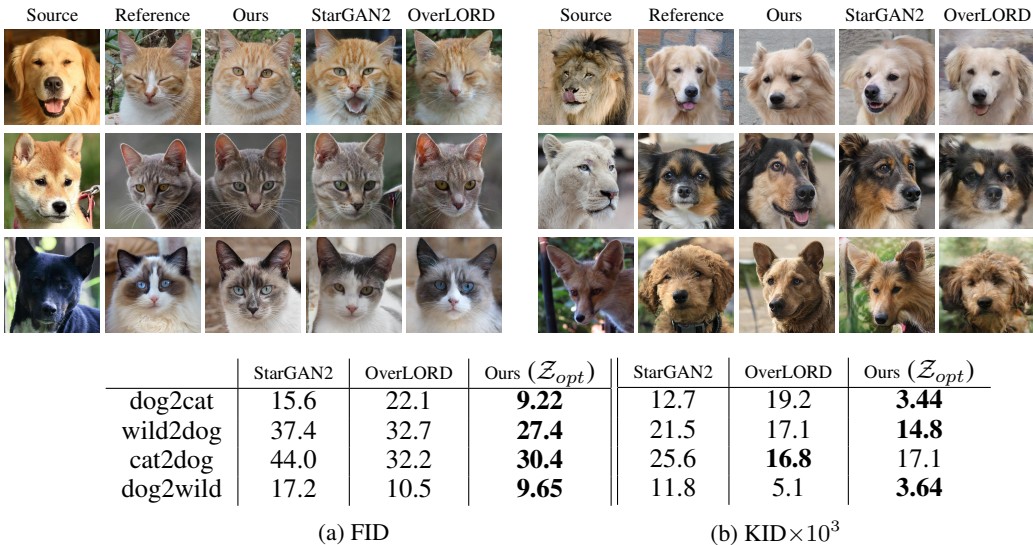

| | StarGAN2 | OverLORD | Ours ($\mathcal{Z}_{opt}$) | StarGAN2 | OverLORD | Ours ($\mathcal{Z}_{opt}$) |
|---|---|---|---|---|---|---|
| dog2cat | 15.6 | 22.1 | **9.22** | 12.7 | 19.2 | **3.44** |
| wild2dog | 37.4 | 32.7 | **27.4** | 21.5 | 17.1 | **14.8** |
| cat2dog | 44.0 | 32.2 | **30.4** | 25.6 | **16.8** | 17.1 |
| dog2wild | 17.2 | 10.5 | **9.65** | 11.8 | 5.1 | **3.64** |
| | (a) FID | | | (b) KID$\times 10^3$ | | |

Figure 6: Comparison of reference-based image translation with StarGAN2 and OverLORD. Our method generates realistic target domain images that combine pose and structure from the source image with texture and color from the reference. StarGAN2 follows the source shape too closely, resulting in non-realistic animals (1st example in dog2cat, all examples in wild2dog). OverLORD's results preserve the appearance of the reference well, but sometimes fail to capture the pose and structure (e.g., ear shape) from the source image (2nd and 3rd examples in wild2dog). A quantitative comparison in the table below indicates superior performance of our method in both FID and KID.

early latent code (below $32 \times 32$ resolution) from a source image, with the late latent code (above or equal to $32 \times 32$ resolution) from a target domain reference image, and feed it to the target model to generate the result, as demonstrated in Figure 22. Figure 23 and Table 6 show that here, as well as for I2I, inversion via $\mathcal{Z}_{opt}$ works better than other inversions/spaces for multi-modal image translation. Figure 6 demonstrates that our results are better than those of current state-of-the-art methods, StarGAN-v2 (Choi et al., 2020) and OverLORD (Gabbay & Hoshen, 2021).

## 4.2 CROSS-DOMAIN IMAGE MORPHING

Image morphing is a popular visual effect of smoothly transitioning between a pair of input images (Wolberg, 1998), which typically requires either manual or automatic correspondences, in order to define a warp field. Cross-domain morphing, where the two images are from different domains, $A$ and $B$, is particularly challenging (Aberman et al., 2018; Fish et al., 2020). However, using a pair of aligned StyleGAN models for the two domains, it is possible to perform cross-domain image morphing automatically without the need for correspondences, or any other input! The two input images are first embedded into the $\mathcal{W}+$ space of the corresponding generators, using e4e encoders (Tov et al., 2021). Next, a smooth transition is obtained by linearly interpolating between the resulting latent codes, while also interpolating between the model weights. Wang et al. (2019) previously proposed interpolating model weights in order to obtain a smooth transition between the "effects" of two different networks. We note that they do not discuss morphing real images, which is a slightly different setting, and requires also interpolating latent codes as we propose here.

*Layer swapping*, proposed by Pinkney & Adler (2020), is an alternative approach to morph between domains. We discuss the differences between the two approaches in Section A.7. Concisely, our proposed method ensures a continuous smooth transition, while layer swapping performs the transition as a series of discrete steps, rather than continuously.

We demonstrate automatic morphing between dog and cat faces in Figures 24 and 25, and dog and human faces in Figures 26 and 27. Interpolating the model weights (along each column) yields a smooth transition between domains (different species, but the same pose and fur color), while interpolating the $\mathcal{W}+$ latent codes (along each row) smoothly transitions inside each domain (same species, varying pose and fur color). In fact, any trajectory in this 2D interpolation space yields a smooth morph sequence between two input images. We simultaneously interpolate along both dimensions to create the sequences shown in the supplementary video.

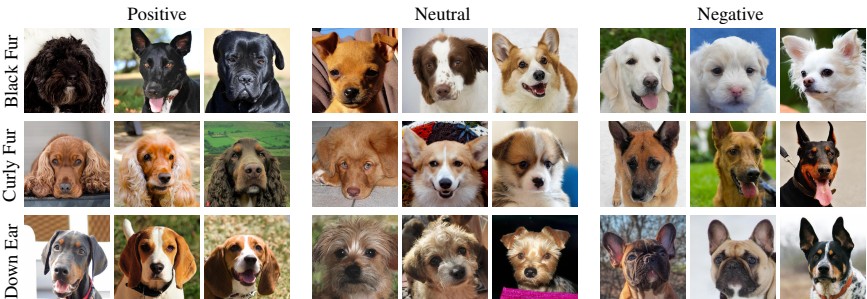

Figure 7: Zero-shot dog attribute classification using aligned models (FFHQ and AFHQ dogs). In the top row a human "black hair" classifier becomes a "black fur" classifier, a "curly hair" classifier is able to classify "curly fur", and a "long hair" classifier becomes a "down-pointing ears" classifier. The neutral columns correspond to images whose prediction scores are close to the cutoff value.

## 4.3 KNOWLEDGE TRANSFER FROM PARENT TO CHILD DOMAIN

Vision tasks on human faces have been researched for years. Consequently, numerous datasets with detailed annotations exist. For example, images in the CelebA dataset (Liu et al., 2015) are labeled with 40 attributes such as "Young", "Curly hair", "Smiling", etc. Such annotations are not available for almost any other domain, such as animal faces, severely limiting the range of tasks that can be solved. As discussed earlier, this issue is a prominent motivation for transfer learning. However, common transfer learning approaches are not applicable in the "zero-shot" setting, where there is abundant labeled data in the source domain, but strictly unlabeled data in the target domain.

We next show that this problem can be solved effectively for directly comparable attributes across domains by leveraging aligned models. Consider the case of head pose (specifically, yaw): a clear and comparable attribute for both humans and dogs, however for humans there is abundant labeled data and for dogs there is none. As demonstrated earlier, the latent pose semantics are aligned in the two models, and the parent's yaw editing direction continues to edit yaw in the child. As shown earlier, this holds for additional attributes. Thus, despite a major gap between the two domains in image space, the gap in the latent space is considerably smaller, enabling transfer of knowledge between these domains. While naïvely applying a model trained on the source images to the target images would fail, this approach works well when applied on the latent representation. To demonstrate this approach, we solve several zero-shot classification and regression tasks using models trained in the latent space of the parent StyleGAN model.

For regression tasks, we use LARGE (Nitzan et al., 2021), which demonstrated that the distance in $\mathcal{W}+$ space to the decision hyperplane associated with a semantic property, gauges the degree of that attribute in image space. See appendix (Section A.6) for more details. Zero-shot yaw regression results are depicted in Figures 28 and 29. As evident, the estimated yaw not only captures the correct tendency, but also produces a value that qualitatively seems reasonably close to actual yaw degree.

For classification tasks we take a similar approach. We simply replace the linear regression model with a logistic regression model and use a cutoff value of 0.5. As shown in Figure 7, our method can turn classifiers for human faces to classifiers for dog faces. These results also demonstrate that the attributes are not required to be exactly identical (pose to pose) but could be comparable in a more broad sense (long hair in humans to down-pointing ears in dogs).

## 5 CONCLUSION

In this work, we performed the first extensive investigation of the properties of *aligned generative models*. We initially answered several open questions, crucial for their understanding. The findings demonstrated impressive and surprising properties, such as semantic alignment across distant domains and knowledge being "hidden" instead of being forgotten. We then leveraged our new insights to apply aligned models for a multitude of tasks. Interestingly, we obtain state-of-the-art results for those tasks with a single, simple fine-tuning based method. We hope that our work can inspire others to consider aligned models as a simple paradigm for solving a wide range of tasks.

## 6 ETHICS STATEMENT

This work performs an extensive study of the properties of *aligned generative models* and applies such models for several computer vision tasks. In general, generative models and learning-based algorithms raise several concerns. Notably, generative models may be used to produce deceiving or offending content, e.g. deepfakes (Wikipedia, 2021), and data-driven algorithms may perpetuate biases exiting in their training sets. However, these concerns are general to the entire fields and are not amplified by this work.

## 7 REPRODUCIBILITY

Throughout the paper we provide detailed information facilitating reproduction of our results. For example, in each experiment we specify the choices of latent space, specific layer and inversion method (e.g., Sections 4.1, 4.2 and A.5). Similarly, when applying latent editing directions we specify with which method were they identified and in what space (e.g., Section 3, A.3). Additionally, we provide in the appendix (Section A.8) the information required to reproduce the child models. We expect these to be sufficient for independent replication of our main findings. Separately, source code and pretrained models have been made available in the project's repository.

## 8 ACKNOWLEDGMENTS

We thank Daniel Cohen-Or for helpful discussions and encouragement and the anonymous reviewers for their comments. This work was supported in part by a gift from Adobe, by the Israel Science Foundation (grant no. 2492/20), and the Joint NSFC-ISF Research Grant Program (3611/21).

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

# A APPENDIX

## A.1 EFFECT OF TUNING ON INDIVIDUAL FEATURE CONVOLUTION LAYERS

As part of our investigation of which parts of the network change during fine-tuning, we also examine in more detail the effect of resetting the weights of individual feature convolution layers on the generated images. We reset one layer at a time and measure the perceptual change in an image using LPIPS. Results are displayed in Table 2. As may be seen, each layer has an effect on the image, but resetting the middle resolution layers (32, 64, 128) induces the greatest LPIPS change.

## A.2 FURTHER SEMANTIC ALIGNMENT ANALYSIS

Since many semantic attributes cannot be controlled by a single style channel, nor by a single manipulation direction in $\mathcal{W}$, we also compare the effect of different semantic manipulation directions in StyleSpace, which are discovered for the parent model using CLIP (Patashnik et al., 2021). Figures 10 and 11 demonstrate that these compound manipulations, e.g., expressions and hair styles, also retain their semantics in the child models.

### A.3 LOCALITY BIAS IN SEMANTICS TRANSFER.

We have demonstrated that a variety of localized controls retain their function during transfer learning between FFHQ and AFHQ. However, since the faces in these two datasets are roughly aligned, it is interesting to examine whether this occurs due to overlap between the corresponding semantic regions. To examine this, we perform transfer learning from a model pretrained on FFHQ (at 256×256 resolution) to three different versions of the same dataset: (i) shifted 60 pixels to the right, (ii) shifted 60 pixels downward, and (iii) flipped upside down. Figure 15 shows that the shifts, and particularly the flip, affect the identity/appearance of the images generated from the same latent codes in $\mathcal{Z}$, however other high-level characteristics, such as gender, age, or hair length, remain similar. We also show the effect of manipulating five different style channels across different layers and different semantic regions. For the horizontally shifted dataset, all five channels retain their function. For the vertical shift, four out of the five channels retain their function (channel 15_45 that controls lipstick loses its effect). For the upside down flip, two out of five channels (9_409 for gaze and 12_479 for blond hair) retain their function. In summary, for 11 out of 15 cases, the function of a channel was transferred despite a significant change in the locality. Thus, locality bias cannot explain all of the alignment that occurs.

There are, however, some interesting examples of strong locality bias. Channel 6_501 controls smiling in the parent FFHQ model, but after an upside-down flip it controls receding hairline, this implies locality bias does contribute to channel-wise semantics transfer, since the forehead of the flipped faces overlaps the mouth location in the original images.

### A.4 SEMANTIC ALIGNMENT BETWEEN DIFFERENT RESOLUTIONS

Given a high resolution StyleGAN2 model, an aligned lower resolution model may be easily obtained by simply removing the high resolution layers, and fine-tuning to convergence. The fine-tuning is necessary, as without it the model generates low-contrast images, as shown in Figure 30. This works well because StyleGAN2 inherently supports multi-resolution synthesis, with the generator containing ToRGB layers and the discriminator containing corresponding FromRGB layers that directly operate in image space for different resolutions. Assuming a high resolution ($1024 \times 1024$) model is already available, creating a low-resolution model ($512 \times 512$) in this way is computationally efficient, requiring less than 2 days of fine-tuning on a single GTX1080Ti GPU, compared to more than one month of training from scratch. Figure 30 shows that the resulting low-resolution model is highly aligned with the original: the same latent code $z \in \mathcal{Z}$ generates nearly the same image, and the semantic controls in the parent model have the same effect in the child model. One of the important consequences of such alignment is that there's no need to spend weeks of GPU time to re-discover the semantic StyleSpace controls (Wu et al., 2020). Furthermore, given an inversion model (Tov et al., 2021) for the parent model, it may be fine-tuned for the child model within a few GPU hours, instead of 2-3 days of training from scratch.

### A.5 METHODS AND SPACES FOR IMAGE TRANSLATION

To determine which latent space and inversion method (encoder or optimization) is best suited for translation of real images, we explore a number of alternatives. We modify the pSp encoder (Richardson et al., 2021) to embed images into $\mathcal{W}$, $\mathcal{Z}$, and $\mathcal{Z}+$ (Song et al., 2021) spaces. For $\mathcal{W}+$ we use the e4e encoder (Tov et al., 2021), which is based on pSp, but generates $\mathcal{W}+$ codes with better alignment with the latent manifold. We also modify the latent optimization method from the official StyleGAN2 implementation (Karras et al., 2020b) to embed into these different spaces. For $\mathcal{Z}/\mathcal{Z}+$, it is crucial to use the truncation trick for both image inversion and generation, otherwise the translation results might exhibit strong artifacts (we use a truncation coefficient of 0.7). Inversion results corresponding to these different methods are shown in Figure 16 for AFHQ dogs and cats.

Examples of I2I translation (dog2wild and cat2dog) using these different inversion methods are shown in Figure 17. While inversion of source domain images to $\mathcal{W}+$ yields arguably the best reconstructions, when translating to the target domain via $\mathcal{W}$ or $\mathcal{W}+$, the color palette of the results seems wrong, especially for the dog2wild translation. We attribute this to the fact that the mapping function (from $\mathcal{Z}$ to $\mathcal{W}$) changes when fine tuning the parent to the child, which affects the color palette, and translating using $\mathcal{W}/\mathcal{W}+$ latent codes ignores this change. Translations via $\mathcal{Z}+$ or $\mathcal{Z}+_{opt}$ inversion also suffer from occasional color artifacts (mainly in the dog2wild examples).

Both $\mathcal{Z}$ and $\mathcal{Z}_{opt}$, on the other hand, yield satisfactory translation results. We prefer $\mathcal{Z}_{opt}$ because it tends to produce a vivid color palette and to maintain a stronger resemblance of the source images, in terms of pose, shape, and colors. Quantitatively, translating via $\mathcal{Z}_{opt}$ inversion achieves best FID and KID over the other plausible alternatives, as reported in Table 5. Therefore we use translation via $\mathcal{Z}_{opt}$ as our preferred method.

We perform similar study for translation between nearby domains (FFHQ and cartoon) in Figure 18, 19 and 20. In our subjective opinion, translation via $\mathcal{Z}_{opt}$ still achieves the most cartoonish look. However, translations via $\mathcal{W}$ bear closer resemblance to the input portrait, while still achieving a satisfactory cartoonish look. As discussed in the text, this may be attributed to the fact that, for similar domains, the mapping function changes little during fine-tuning, resulting in pointwise alignment of the $\mathcal{W}$ spaces of the parent and child models.

## A.6 ZERO-SHOT REGRESSION

To leverage aligned models for regression tasks, we use LARGE (Nitzan et al., 2021), which demonstrated that the distance in $\mathcal{W}+$ space to the decision hyperplane associated with a semantic property, gauges the degree of that attribute in image space. As their method is designed for a few-shot setting, we simplify it slightly for our setting where the training data in the source domain is abundant. Concisely, we simply use the distances calculated in specific layers known to control certain attributes as the input features for the regression model. We demonstrate this approach for head pose regression and use the first four layers, which are known to control the pose in StyleGAN (Karras et al., 2019; Nitzan et al., 2021). At inference time, we use e4e (Tov et al., 2021) to encode images of the target domain into the $\mathcal{W}+$ space of the child model, compute the distances of the first four layers from the decision hyperplane, and input them to the human face yaw estimation model.

The zero-shot yaw regression results for AFHQ dogs and cats are depicted in Figures 28 and 29. As can be seen, the estimated yaw not only captures the correct tendency, but also produces a value that qualitatively seems reasonably close to actual yaw degree.

## A.7 METHODS TO BLEND ALIGNED MODELS

*Layer swapping* was introduced by Pinkney & Adler (2020) as a method to generate images of a new domain by "blending" together two existing data domains. It does that by creating a hybrid model contains layers from two aligned models. Specifically, the first (coarse) layers are taken from one model and the last (fine) layers are taken from another. We note that this method blends the two data domains in a specific manner. Thanks to the hierarchical structure of StyleGAN (Karras et al., 2019), the created model inherits the structure (coarse layers) from one model and texture from another (fine layers). Pinkney & Adler (2020) also mentioned that the fine layers could be interpolated between the models, however this idea wasn't applied in practice.

The layer swapping method was shown to produce visually pleasing results on the task of stylizing human portraits (Pinkney & Adler, 2020; Song et al., 2021). However, there are a few disadvantages to this method. First, when domains are more distant (e.g. faces of humans and dogs), the results obtained by this approach are less intuitive and visually pleasing (see Figures 31 to 33). This coincides well with our observation from Figure 8. Since the feature convolution layers change much more significantly when transferring to a distant domain, the layers of a layer-swapped model are more alien to each other. Second, the number of intermediate steps is limited by the number of convolution layers in the generator, which is at most 18. This prevents the application of layer swapping for creating a smooth transition between images from different domains.

In our morphing application (Section 4.2), we present an alternative method to blend two aligned models. There we propose to perform a simple linear interpolation of all model weights to achieve a gradual transition. We compare the results of this approach with layer swapping in Figures 31 to 33. Please note that our proposed method is able to obtain "blended" images that seem more smooth and natural.

### A.8 FINE-TUNING IMPLEMENTATION DETAILS

Given a model pretrained on the parent domain, we fine-tune it on the child domain. Specifically, we use model config-f and the default hyper-parameters from the official Nvidia StyleGAN2 and StyleGAN2-ADA implementations in tensorflow. We use the augmentations of StyleGAN2-ADA only when the child domain is AFHQ or Metface.

Note that StyleGAN2-ADA implementation chooses the config based on input image resolution. It uses config-f for image resolution above $512 \times 512$, and config-e for other resolutions. To use config-f without worrying about image resolution, one can specify the flag `--cfg stylegan2` when using tran.py, and change line 179 in *train.py* from `spec.fmaps = 1 if res >= 512 else 0.5` to `spec.fmaps = 1`.

### A.9 DETECTING LOCALIZED CHANNELS

We follow Wu et al. (2020) to discover localized channels in the StyleGAN model. To reduce noise, we only consider channels to be localized if they have the strongest gradient in the same semantic region over $75\%$ of sampling images, rather than $50\%$ used in the original paper. For FFHQ and Metface models, we use the semantic segmentation maps from BiSeNet (Yu et al., 2018) pretrained on CelebAMask-HQ (Lee et al., 2020a). For AFHQ dogs, we using the semantic segmentation maps from a unified parsing network (Xiao et al., 2018) pretrained on Broden+ (Bau et al., 2017).

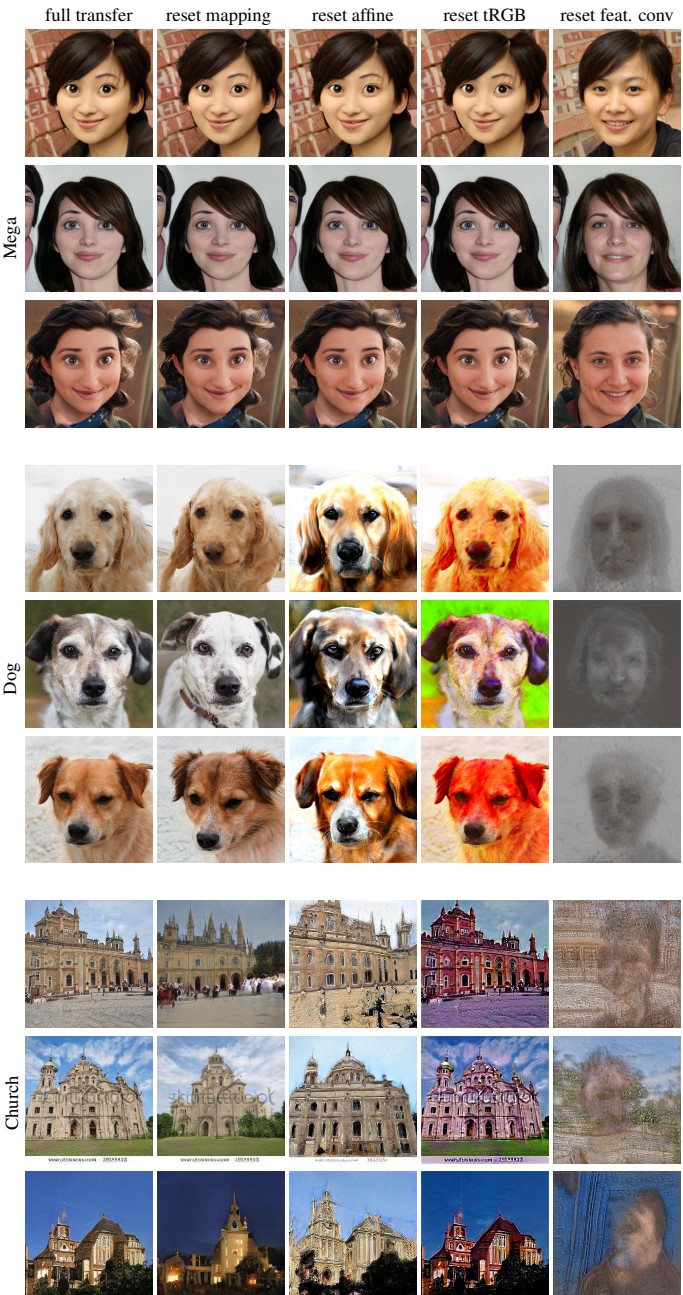

Figure 8: We reset the weights of different components in child models (Mega, dog, church) to their initial values, which come from the parent model (FFHQ). When resetting the weights in feature convolution layers, the output images change more drastically (content, structure), while resetting the weights of other components causes milder effects. This implies feature convolution layers contain most of new learned knowledge.

| Resolution | 4 | 8 | 16 | 32 | 64 | 128 | 256 | 512 |
|---|---|---|---|---|---|---|---|---|
| LPIPS | 0.156 | 0.385 | 0.390 | 0.432 | 0.440 | 0.405 | 0.369 | 0.355 |

Table 2: We measure the extent to which the change in each feature convolution layer (during fine-tuning) affects the generated images. Given a parent FFHQ model and a child AFHQ dog model, we reset the feature convolution weights for each resolution of the child model to their original values in the parent model, and measure the LPIPS distance between the images generated by child model before and after resetting the weights. A higher LPIPS score indicates a more significant change in image space. It may be seen that the greatest change is caused by resetting the middle resolution layers (32, 64, 128).

| | | eyebrow 19 | eye 5 | ear 41 | nose 21 | mouth 32 | neck 46 | cloth 34 | hair 62 |
|---|---|---|---|---|---|---|---|---|---|
| eyebrow | 29 | 8 | | | 1 | | 1 | | |
| eye | 9 | | 3 | | | | | | |
| ear | 45 | | | 20 | | | | | |
| nose | 23 | | | | 8 | | | | |
| mouth | 55 | | | | | 11 | | | |
| neck | 61 | | | | 1 | | 15 | | |
| cloth | 65 | | | | | | | 19 | |
| hair | 70 | | | | | | | | 33 |

Table 3: The number of localized StyleSpace controls for various semantic regions for an FFHQ parent model and an FFHQ grandchild model, with training flow from FFHQ (parent) to AFHQ dog (child) then back to FFHQ (grandchild). Each column corresponds to a semantic region for parent and each row to a semantic region for grandchild. The number of localized channels shared between two models is indicated for each pair of semantic regions.

| | | eyebrow 19 | eye 5 | ear 41 | nose 21 | mouth 32 | neck 46 | cloth 34 | hair 62 |
|---|---|---|---|---|---|---|---|---|---|
| eyebrow | 22 | | | | | 1 | 1 | | |
| eye | 5 | | | | | | | | 1 |
| ear | 44 | | | | | 1 | 1 | 1 | |
| nose | 19 | | | | | | | | |
| mouth | 28 | | | 1 | 1 | 1 | | | |
| neck | 43 | | | 1 | | 1 | 1 | | 1 |
| cloth | 32 | | | 2 | 1 | | | 1 | |
| hair | 85 | | | | | 1 | 2 | | 2 |

Table 4: The number of localized StyleSpace controls for various semantic regions for two randomly initialized FFHQ models. Each column corresponds to a semantic region in one model and each row to a semantic region in the other model. The number of localized channels shared between two models is indicated for each pair of semantic regions. It is evident that the two models only have a small number of overlap channels across unrelated semantic regions (for example, hair and eye). This experiment serves as a negative control to show that a large number of overlap channels only occurs when the two models have parent and child relation, as is the case in Table 1

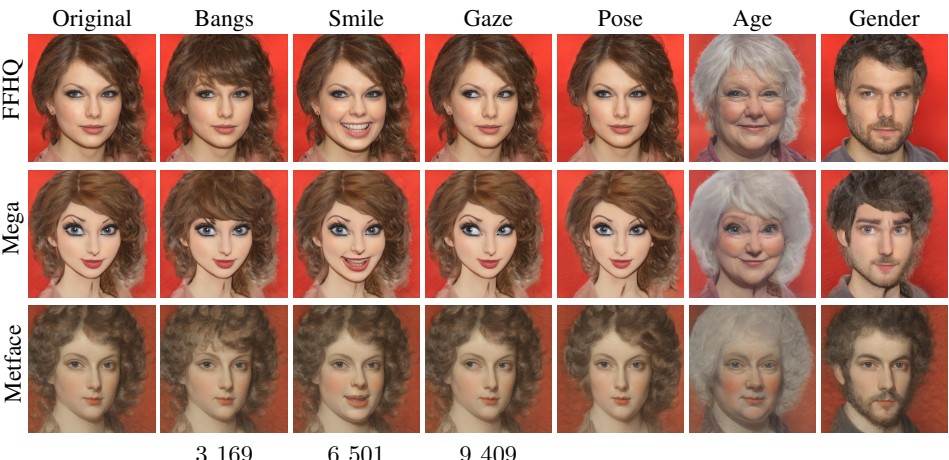

Figure 9: Semantic alignment: semantic controls discovered for the parent model (FFHQ) retain their function in the children models (Mega and Metface). This holds for individual channels in $\mathcal{S}$ (bangs, smile, gaze), where the layer and channel number is indicated under each column. Semantic alignment is also observed for manipulation directions in $\mathcal{W}$ (pose, age, gender).

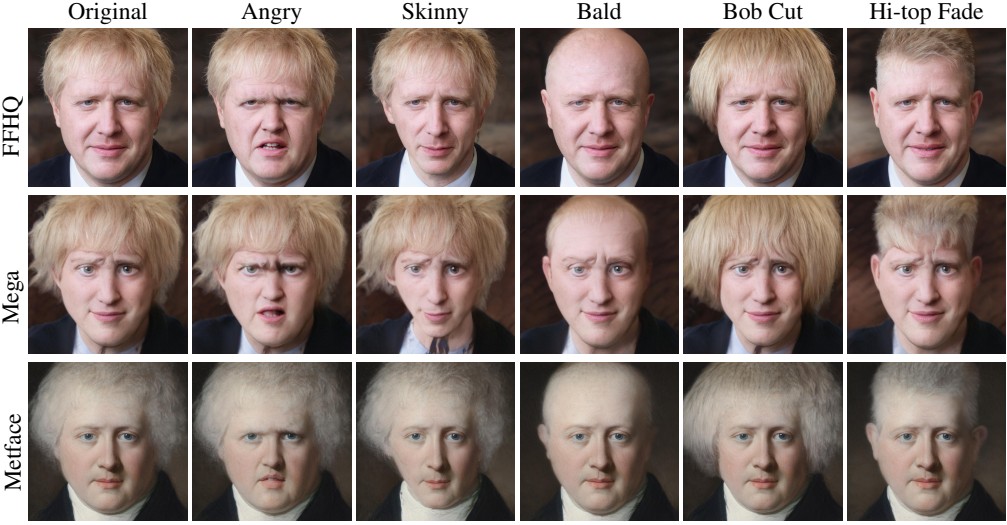

Figure 10: Semantic alignment of multiple channels: semantically meaningful directions in StyleSpace discovered in the parent model (FFHQ), detected using StyleCLIP (Patashnik et al., 2021), still control the same attributes in children models (Mega and Metface).

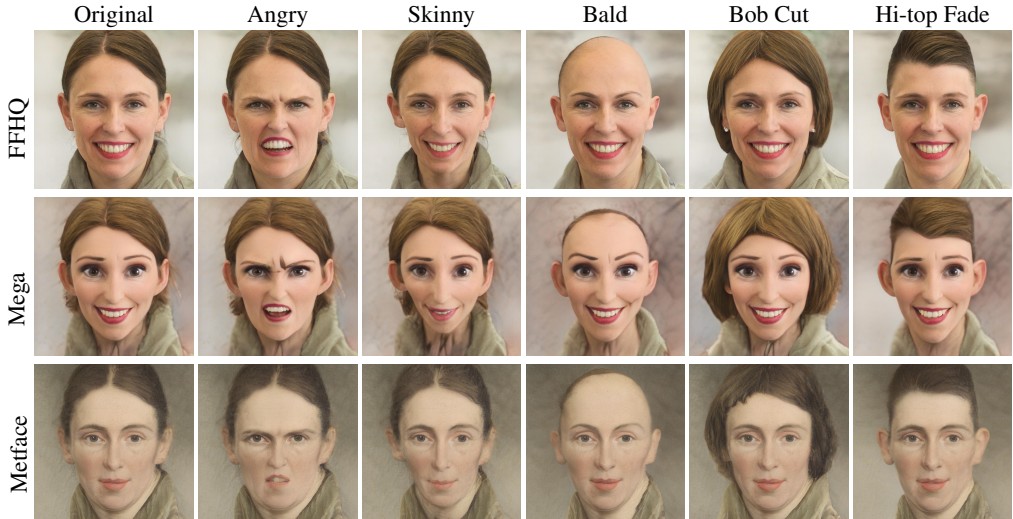

Figure 11: Semantic alignment of multiple channels: semantically meaningful directions in StyleSpace discovered in the parent model (FFHQ), detected using StyleCLIP (Patashnik et al., 2021), still control the same attributes in children models (Mega and Metface).

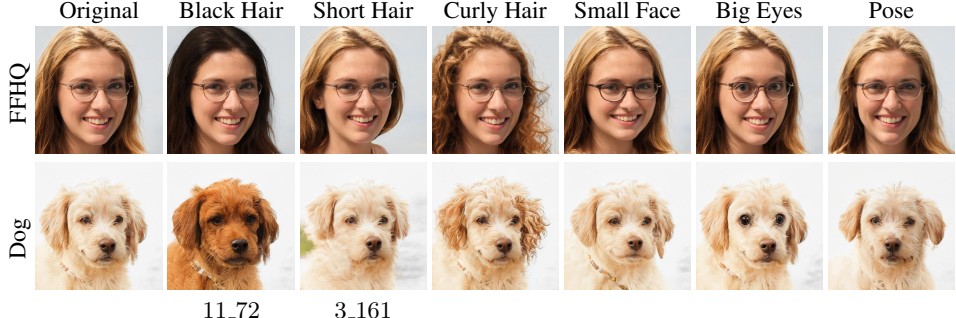

Figure 12: Examples of semantic alignment between single-channel, as well as multi-channel controls discovered for the parent model (StyleGAN2 trained on FFHQ) and a child model (AFHQ dogs). While the analogy between hair in humans and fur in dogs seems intuitive, there are also some less obvious analogies, such as hair length and ear length.

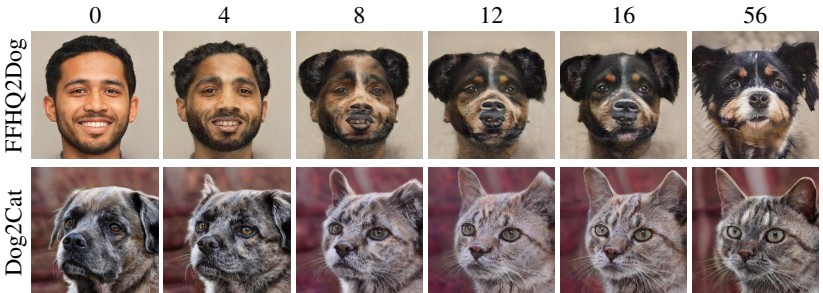

Figure 13: During transfer learning between domains, we can observe a smooth transition in images generated from the same latent code $z \in \mathcal{Z}$. The top row demonstrates this for transfer from FFHQ to AFHQ dogs, while the bottom rows shows this for transfer from AFHQ dogs to cats. The number of epochs is indicated above each column. The most significant visual changes occur in early epochs (0–16), while later epochs mainly improve image quality and realism without significant changes in semantic attributes.

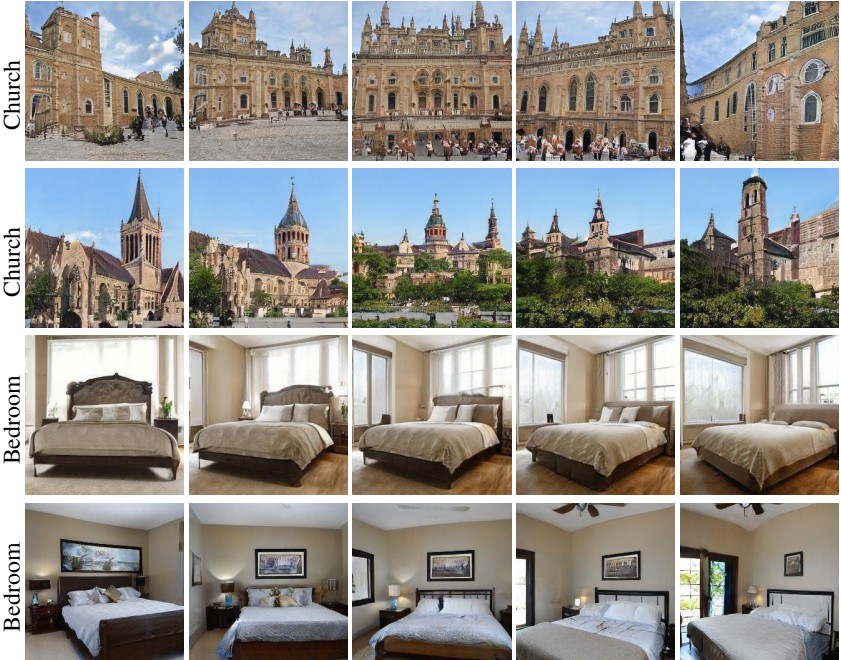

Figure 14: Some degree of semantic alignment is present even when the source and target domains are very dissimilar. In the top two rows, we show that the latent direction that controls pose in the parent FFHQ model still controls pose in the child LSUN church model. In the bottom two rows, we examine a double transfer, with FFHQ as parent, AFHQ dog as child and LSUN bedroom as grandchild. The pose direction in FFHQ still controls the pose in the grandchild bedroom model.

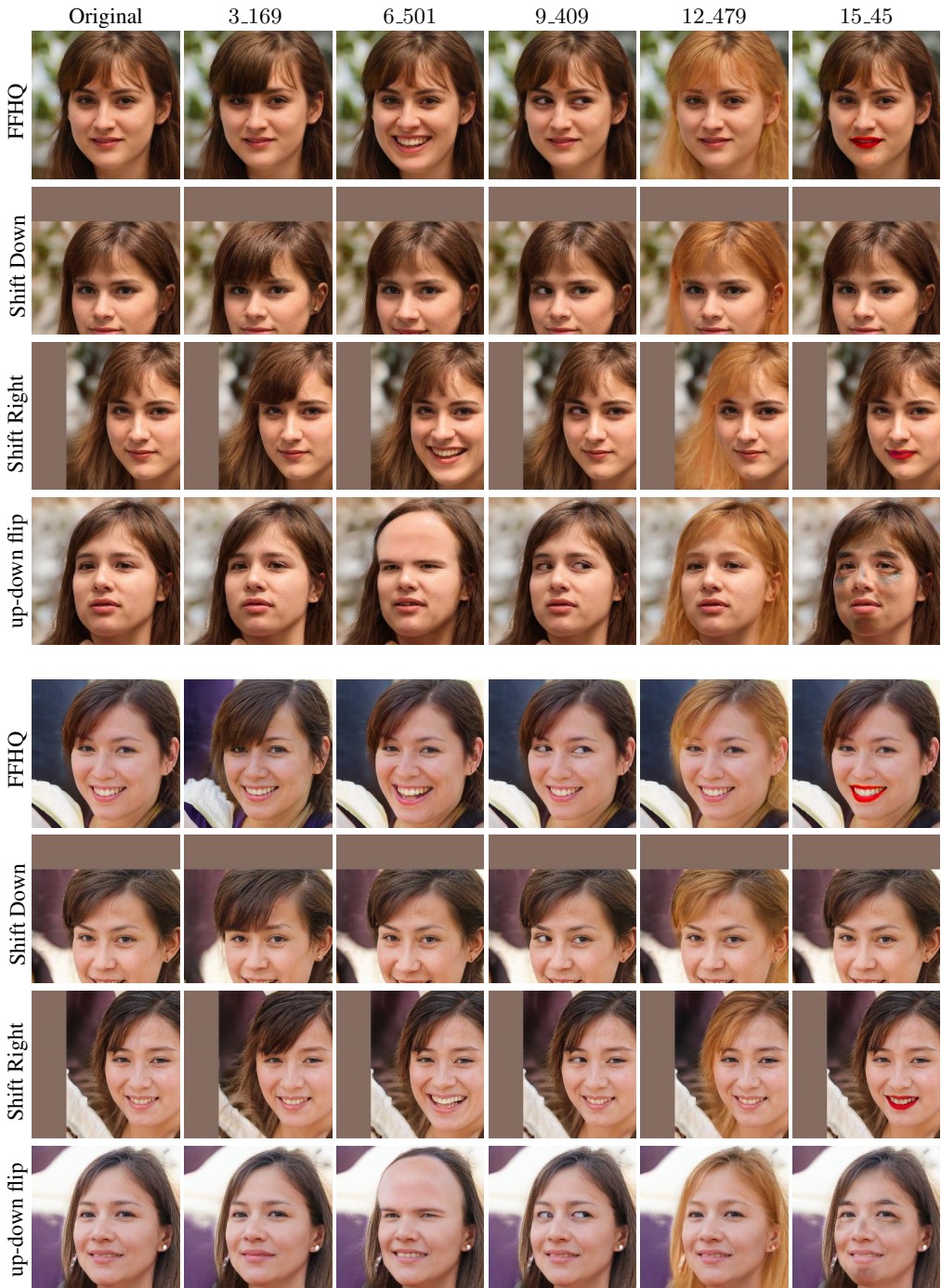

Figure 15: To understand whether locality bias contributes to semantics transfer, we fine-tune a pretrained FFHQ model in 256×256 resolution, to (i) a FFHQ dataset shifted 60 pixels to the right, (ii) a FFHQ dataset shifted 60 pixels downward, and (iii) a FFHQ dataset flipped upside-down. We examine the semantics transfer for 5 channels across different layers and different semantic regions. For the shift right case, all 5 channels retain their function. For the shift down case, 4 out of 5 channels retain their function (channel 15_45 loses its function for lipstick). For the upside-down flip, 2 out of 5 (9_409 gaze and 12_479 blond hair) retain their function. In summary, for 11 out of 15 cases, the semantic function of channels is transferred even if we break the locality bias. These results imply that the transfer of semantics cannot be fully attributed to locality bias.

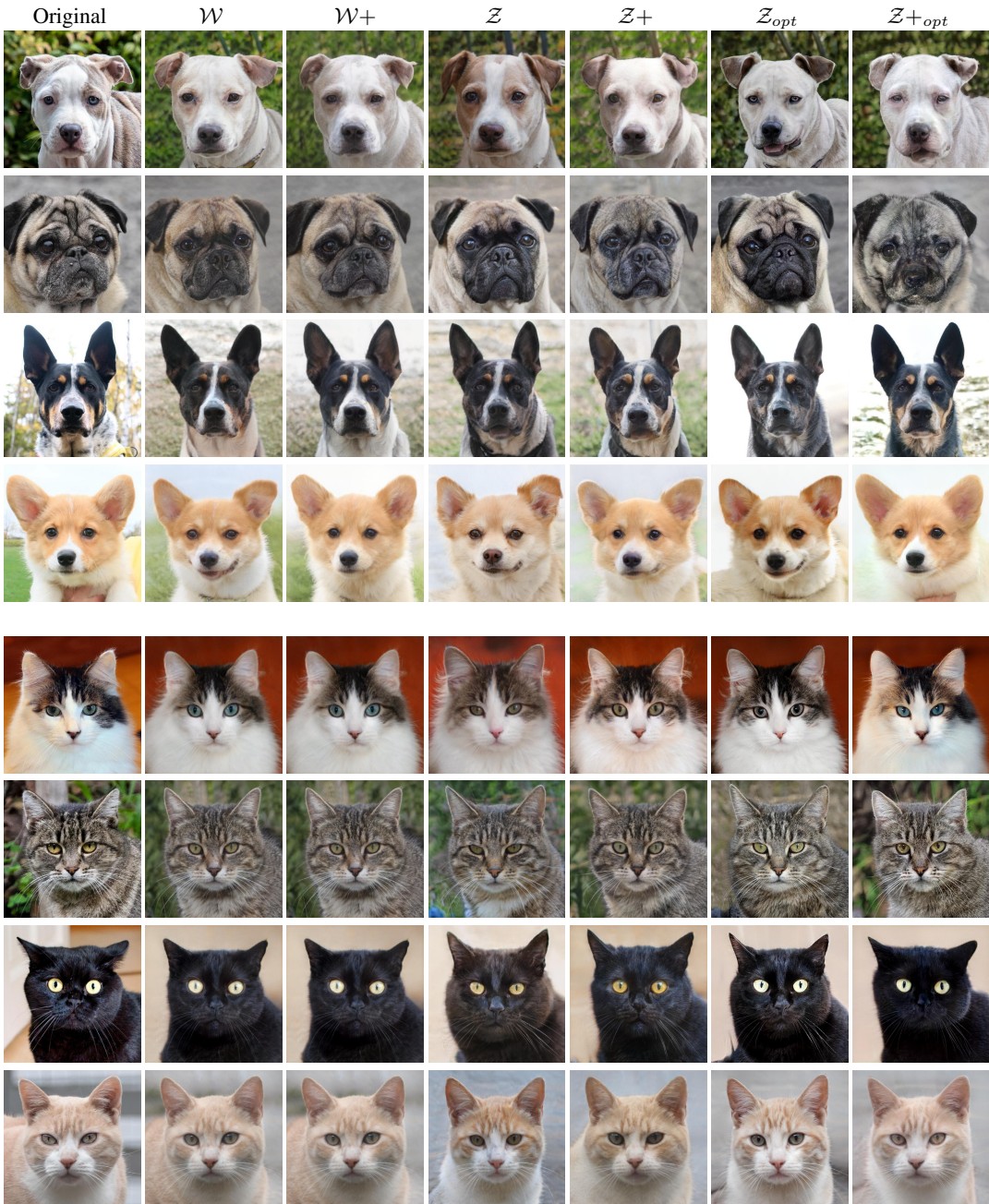

Figure 16: To invert real images of animal faces to different latent spaces, we examine both encoders and latent optimization based methods. We use the pSp encoder (Richardson et al., 2021) as a backbone and modify it to embed into $\mathcal{W}$, $\mathcal{Z}$, and $\mathcal{Z}+$ (Song et al., 2021) spaces. For the $\mathcal{W}+$ space, we use e4e (Tov et al., 2021), which also uses pSp (Richardson et al., 2021) as backbone. For optimization based inversion, we modify the optimization code from StyleGAN2 (Karras et al., 2020b) to $\mathcal{Z}$ or $\mathcal{Z}+$ space (two rightmost columns). All of the inversion methods yield reasonably faithful reconstructions, with occasional artifacts in the $\mathcal{Z}$ and $\mathcal{Z}+_{opt}$ reconstructions. Note that, as we show below, that better reconstruction does not necessarily yield the best image translation.

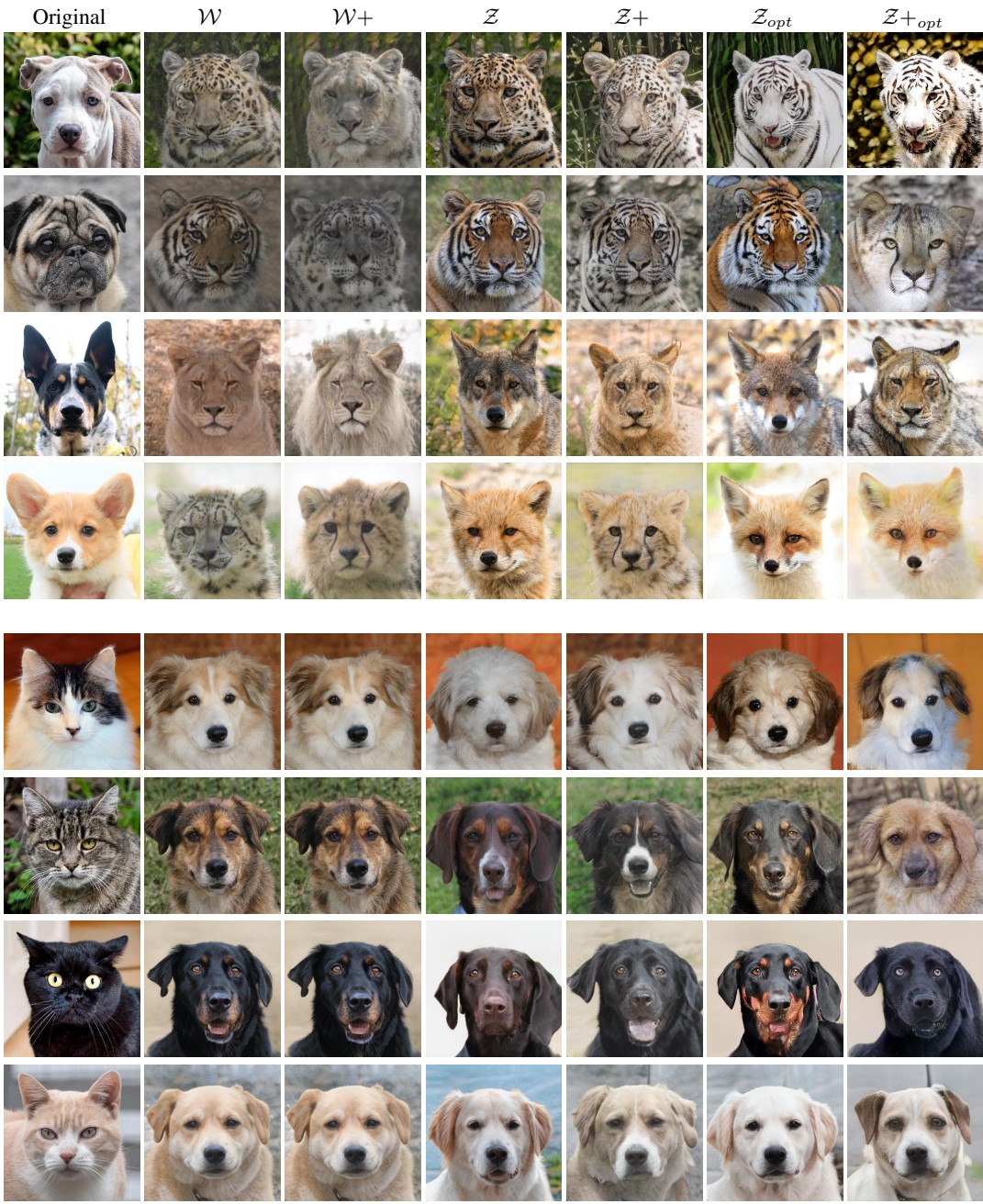

Figure 17: Comparison of I2I results (dog2wild in the top four rows, cat2dog in the four bottom ones) for the different inversions shown in Figure 16. The color palette appears to be wrong for both $\mathcal{W}+$ and $\mathcal{W}$ encoding, especially for the dog to wildlife translation. This is not surprising, since the mapping function changes during fine tuning (see Figure 1), affecting the color palette, and inverting into the $\mathcal{W}$ or $\mathcal{W}+$ spaces ignores the difference between the mapping functions of the parent and child. Translations via $\mathcal{Z}+$ or $\mathcal{Z}+_{opt}$ inversion also suffer from occasional color artifacts (mainly in the dog2wild examples). Translations via either $\mathcal{Z}$ or $\mathcal{Z}_{opt}$ provide satisfactory results. We prefer $\mathcal{Z}_{opt}$ because it typically yields a more vivid color palette, while slightly better capturing the characteristics of the source images (especially in the dog2wild examples).

Original    $\mathcal{W}$    $\mathcal{W}+$    $\mathcal{W}_{opt}$    $\mathcal{W}+_{opt}$    $\mathcal{Z}_{opt}$    $\mathcal{Z}+_{opt}$

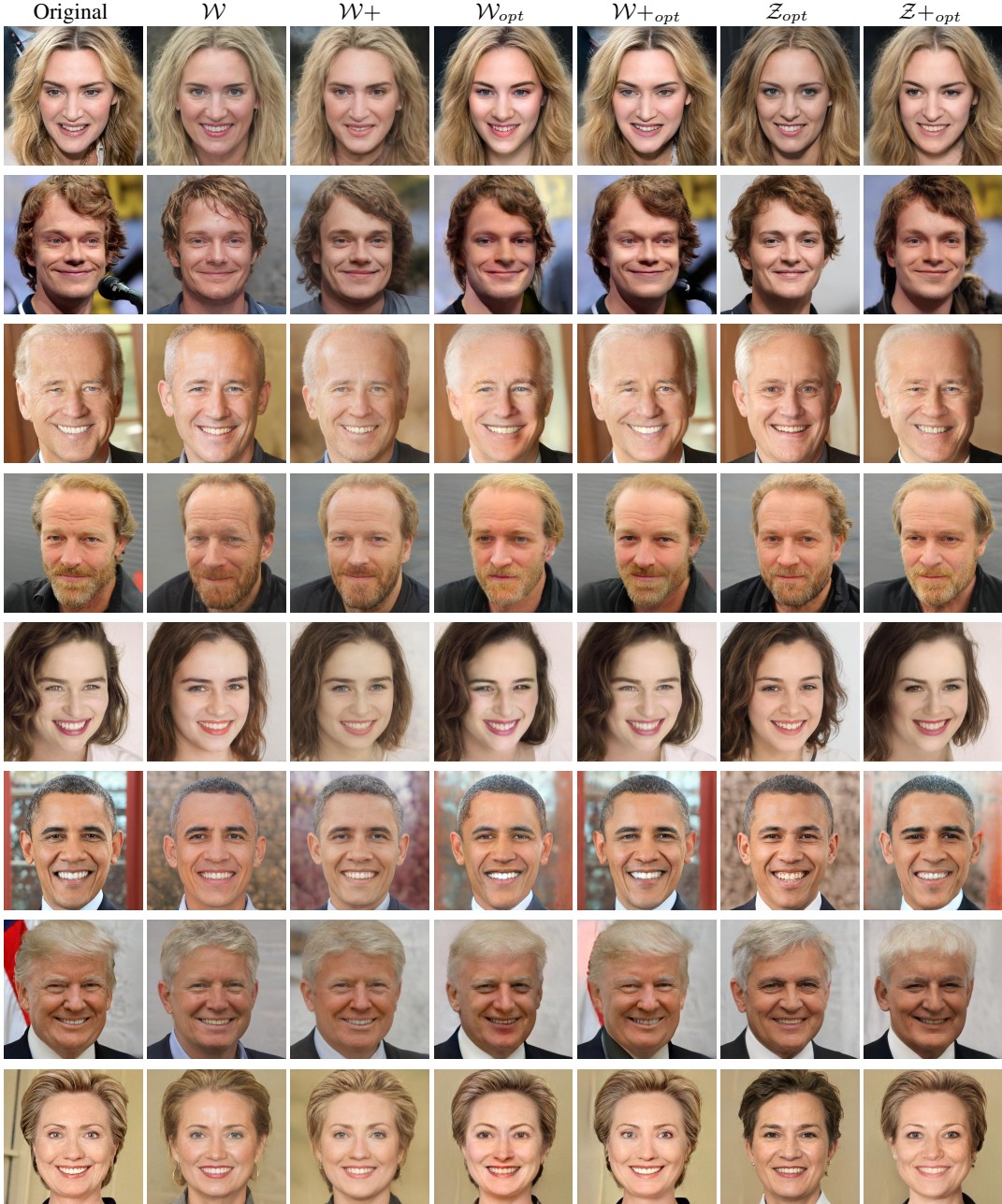

Figure 18: To invert real images of human faces to different latent spaces, we examine both encoders and latent optimization based methods. We use the pSp encoder (Richardson et al., 2021) as a backbone and modify it to embed into $\mathcal{W}$ space. For the $\mathcal{W}+$ space, we use e4e (Tov et al., 2021), which also uses pSp (Richardson et al., 2021) as backbone. We also experimented with using the pSp encoder to $\mathcal{Z}$, and $\mathcal{Z}+$ (Song et al., 2021) spaces, but training does not converge and results are unrealistic. For optimization-based inversion, we modify the optimization code from StyleGAN2 (Karras et al., 2020b) to $\mathcal{W}$, $\mathcal{W}+$, $\mathcal{Z}$ or $\mathcal{Z}+$ spaces. In terms of reconstruction quality alone, $\mathcal{W}+$ typically yields the best inversions; however, as we show below, better reconstruction does not necessarily yield the best image translation.

Original     $\mathcal{W}$     $\mathcal{W}+$     $\mathcal{W}_{opt}$     $\mathcal{W}+_{opt}$     $\mathcal{Z}_{opt}$     $\mathcal{Z}+_{opt}$

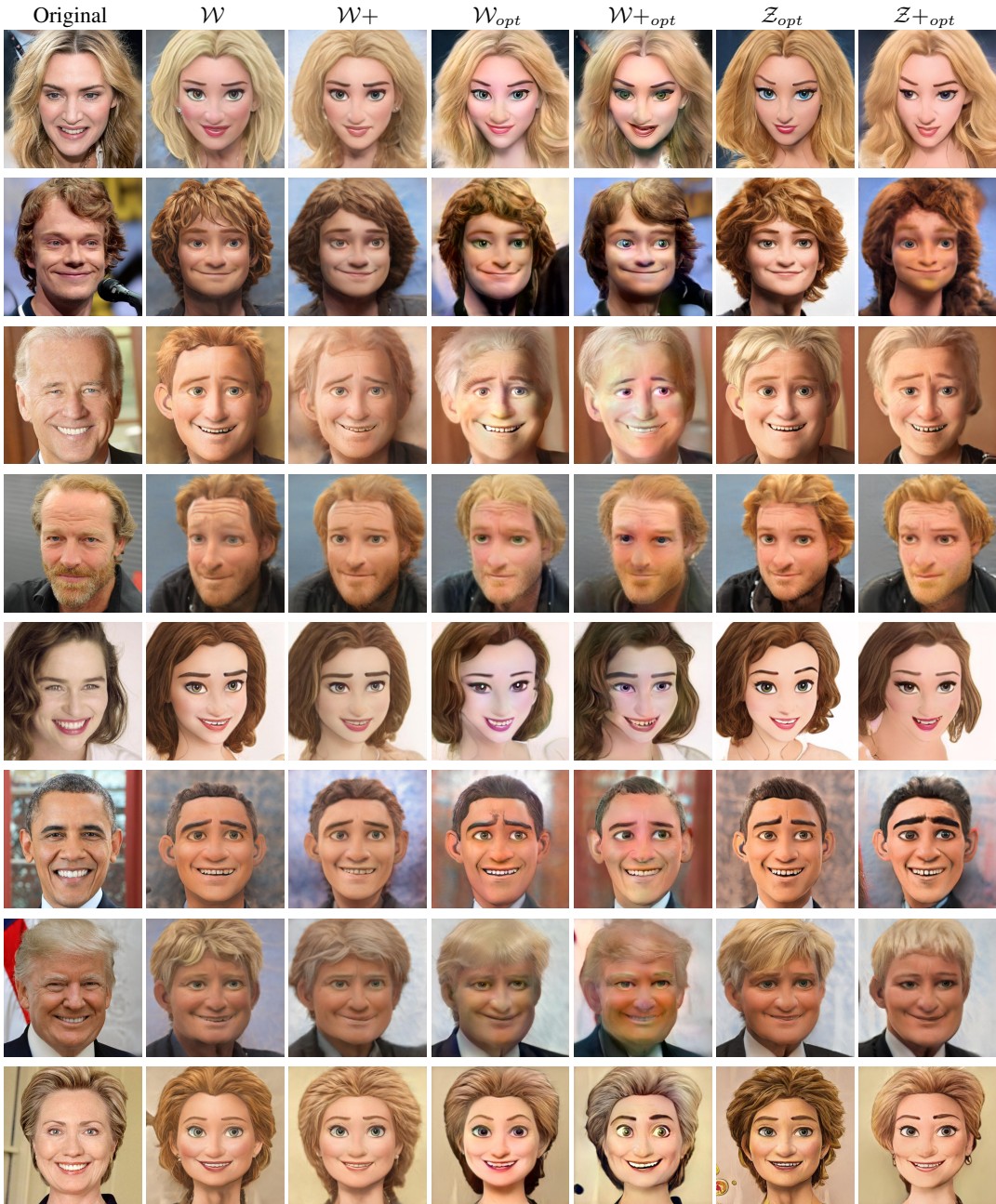

Figure 19: Comparison of I2I results (for real faces to cartoon-like, using FFHQ parent and Mega child) for the different inversions shown in Figure 18. Translation results via $\mathcal{W}_{opt}$ and $\mathcal{W}+_{opt}$ contain strong artifacts. In our subjective opinion, translation via $\mathcal{Z}_{opt}$ achieves the most cartoonish look. However, translations via $\mathcal{W}$ bear closer resemblance to the input portrait, while still achieving a satisfactory cartoonish look. As discussed in the text, this may be attributed to the fact that, for similar domains, the mapping function changes little during fine-tuning, resulting in pointwise alignment of the $\mathcal{W}$ spaces of the parent and child models.

|         | $\mathcal{Z}_{enc}$ | $\mathcal{Z}+_{enc}$ | $\mathcal{Z}_{opt}$ | $\mathcal{Z}_{enc}$ | $\mathcal{Z}+_{enc}$ | $\mathcal{Z}_{opt}$ |
|---------|------|------|----------|------|------|----------|
| cat2dog | 48.8 | 68.5 | **34.2** | 16.1 | 34.8 | **7.36** |
| dog2wild | 22.1 | 24.8 | **10.9** | 10.9 | 12.2 | **2.19** |
| wild2dog | 60.0 | 62.5 | **34.7** | 15.5 | 22.7 | **5.98** |
| dog2cat | 30.4 | 21.1 | **17.9** | 14.5 | 5.49 | **3.79** |
|         | (a) FID | | | (b) KID$\times 10^3$ | | |

Table 5: A quantitative comparison of I2I translation via different latent spaces and inversion methods. Based on the qualitative results shown in Figure 17, we consider encoder-based inversion for $\mathcal{Z}$ and $\mathcal{Z}+$ spaces, and latent optimization method for $\mathcal{Z}$ space, to be promising methods and further examine them using FID and KID scores. Our results indicate that inversion $\mathcal{Z}$ using latent optimization achieves the best FID and KID for I2I translation tasks.

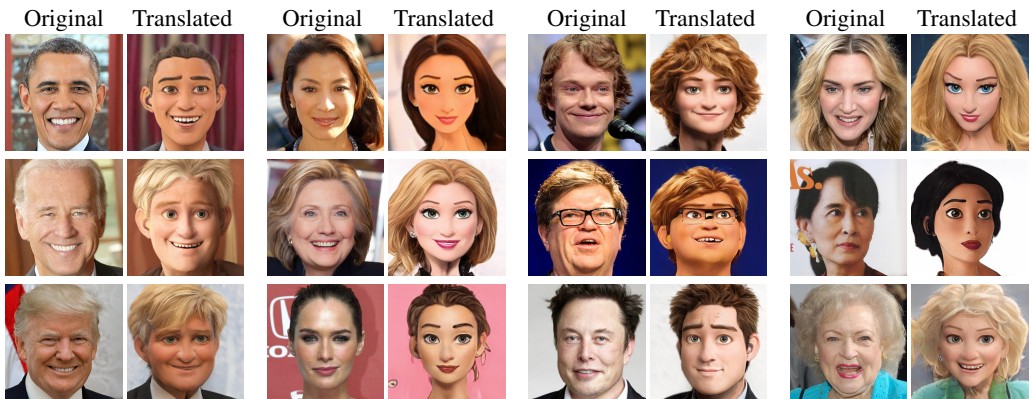

Figure 20: Image Toonification using our $\mathcal{Z}_{opt}$ method.

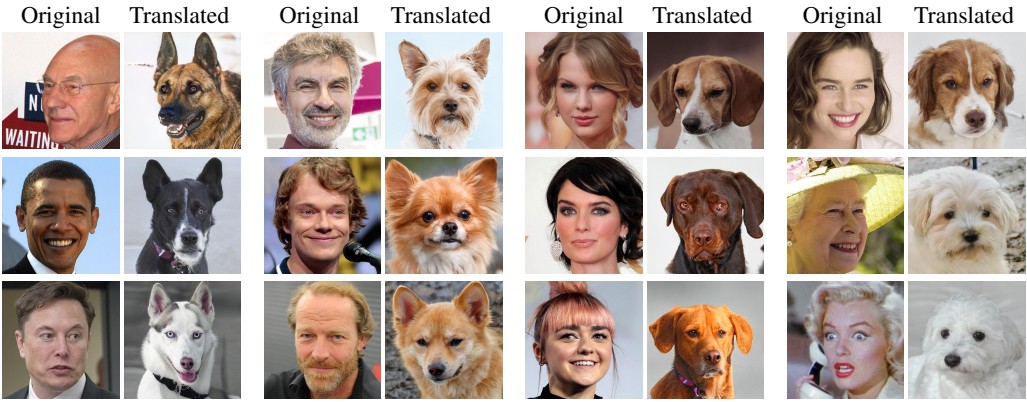

Figure 21: Aligned models enable effective image translation between dissimilar domains (human face and dog face). Some interesting analogies emerge in these translations. For example, as the human hair becomes longer, so does the dog's fur, while the dog's ears change from "candle flame" ears, to "bat" ears, and finally to folded ("down-pointing") ears. The fur color is mainly determined by the human hair color, and the dog pose mimics that of the human.

| Source | Reference | 0 | 3 | 6 | 9 | 12 | 15 | 18 | 21 | 23 |

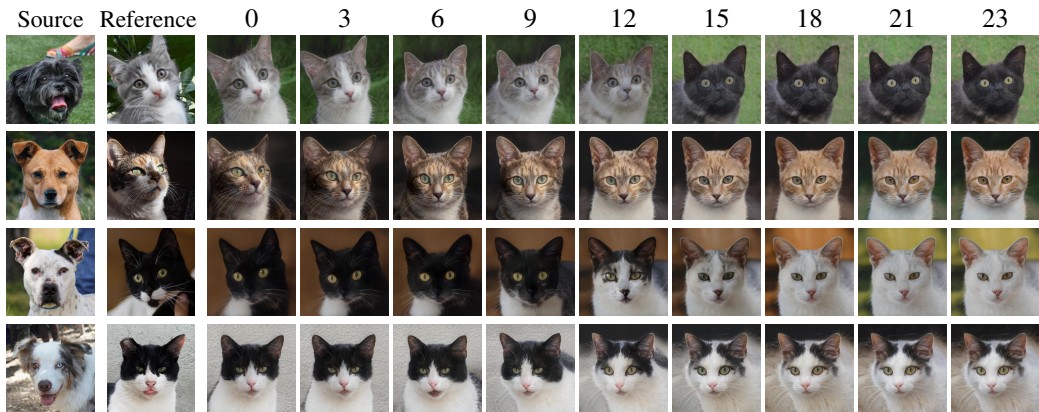

Figure 22: Reference-based image translation. Given a real dog image as source and a real cat image as reference, we aim to obtain a cat image that keeps the content (mainly pose) from the source and the style (fur texture and color) from the reference. We first invert the input real images to latent space of StyleGAN, then take style codes for all layers below $n$ (low resolution) from the source, and style codes for layers above or equal to $n$ (high resolution) from the reference. The layer index $n$ is indicated above each column. Thus, index 0 represents the inverted reference, and index 23 represents the translation of the source to the target domain (cats), while the other indices correspond to standard style mixing in StyleGAN. We can see that when $n$ is around 6, the images combine the pose of the source with the style of the reference.

|  | w+ enc | Z+ enc | Z enc | z_opts | w+ enc | Z+ enc | Z enc | z_opts |
|---|---|---|---|---|---|---|---|---|
| dog2cat | 10.3 | 11.2 | 13.7 | **9.22** | 4.87 | 4.85 | 6.56 | **3.43** |
| wild2dog | 44.5 | 37.6 | 36.6 | **27.4** | 27.2 | 21.1 | 18.8 | **14.8** |
| cat2dog | 42.1 | 44.3 | 40.7 | **30.4** | 27.6 | 28.2 | 27.0 | **17.0** |
| dog2wild | 37.9 | 28.3 | 18.5 | **9.65** | 21.5 | 16.2 | 12.5 | **3.64** |

(a) FID  (b) KID$\times 10^3$

Table 6: A quantitative comparison of reference-based image translation using different inversion methods and latent spaces. It may be seen that the latent optimization method for $\mathcal{Z}$ achieves the best FID and KID for such translation tasks.

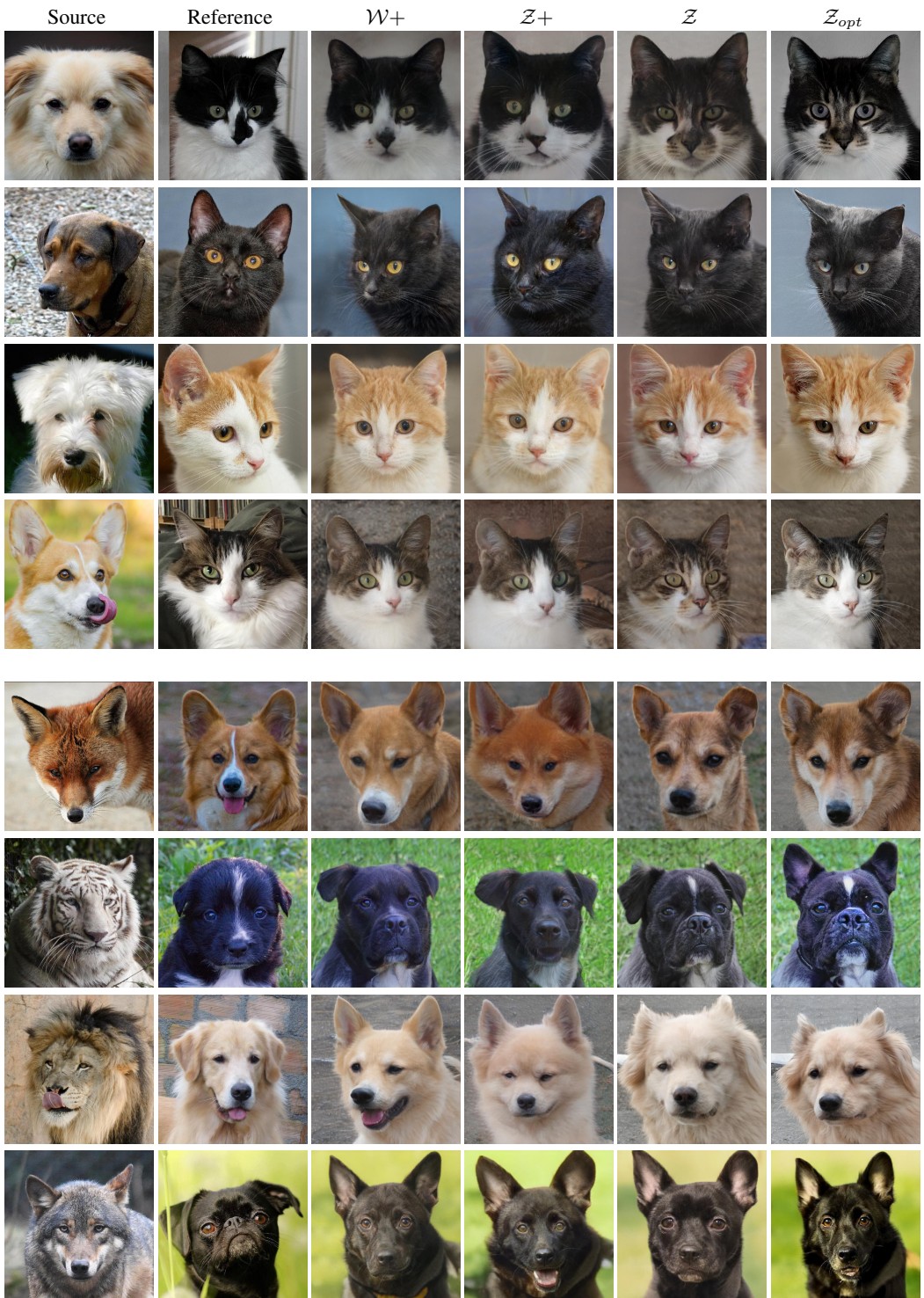

| Source | Reference | $\mathcal{W}+$ | $\mathcal{Z}+$ | $\mathcal{Z}$ | $\mathcal{Z}_{opt}$ |

Figure 23: A qualitative comparison of reference-based image translation for different methods and spaces. Since here the colors are determined by the higher layers of the generator, whose style parameters come from the inversion of the reference image, the translation via $\mathcal{W}+$ does not suffer from color palette issues. Thus, both translations via $\mathcal{W}+$ and via $\mathcal{Z}_{opt}$ look satisfactory.

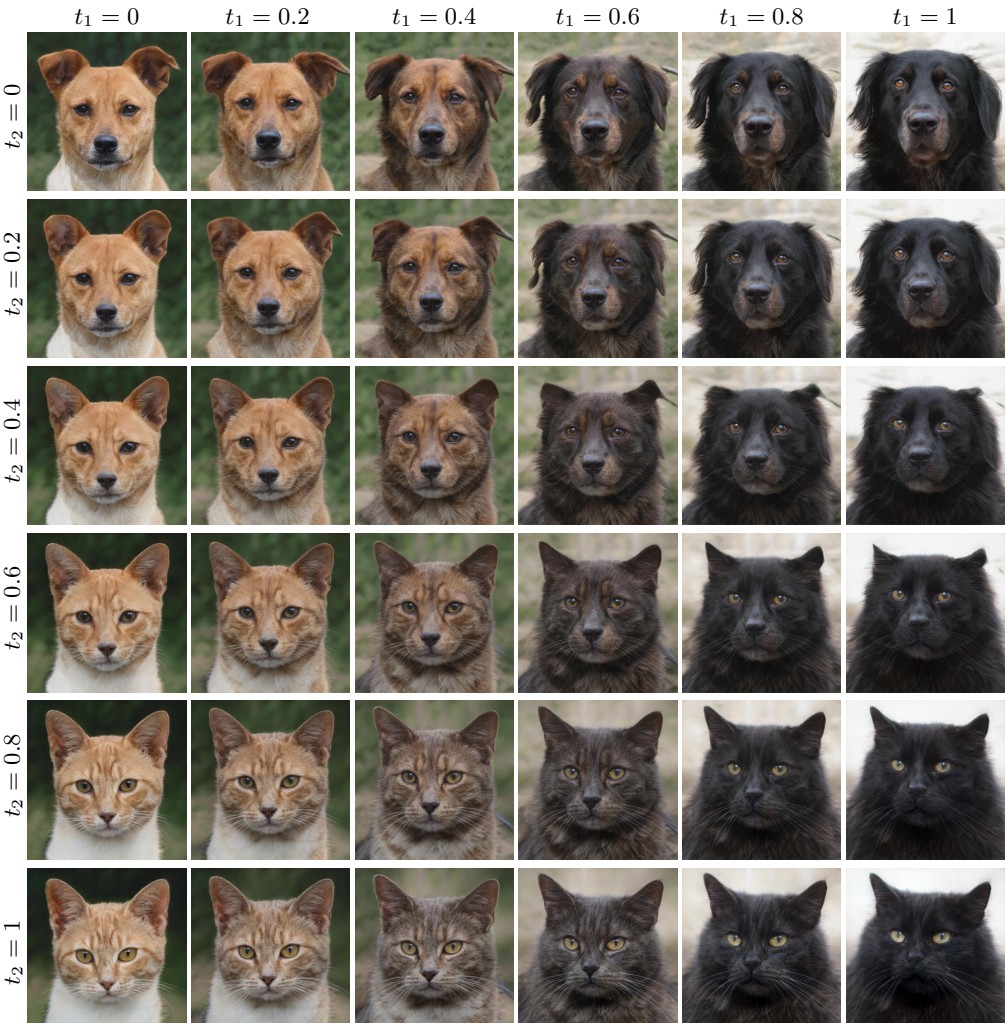

Figure 24: Given a pair of real images from domain $A$ (top-left) and $B$ (bottom-right), we smoothly transition between them by interpolating their latent codes in $\mathcal{W}+$, as well as the model weights. $t_1$ is the interpolation coefficient for the latent codes, while $t_2$ is the coefficient for the model weights. In the same column (fixed $t_1$), we obtain a smooth transition between the domains (different species, but the same pose and fur color). In the same row (fixed $t_2$), we have a smooth transition inside the same domain (same species, varying pose and fur color). Any trajectory between the top-left and bottom-right corners yields a smooth morph sequence between two input images. See the accompanying video, which progresses along the diagonal $t_1 = t_2$.

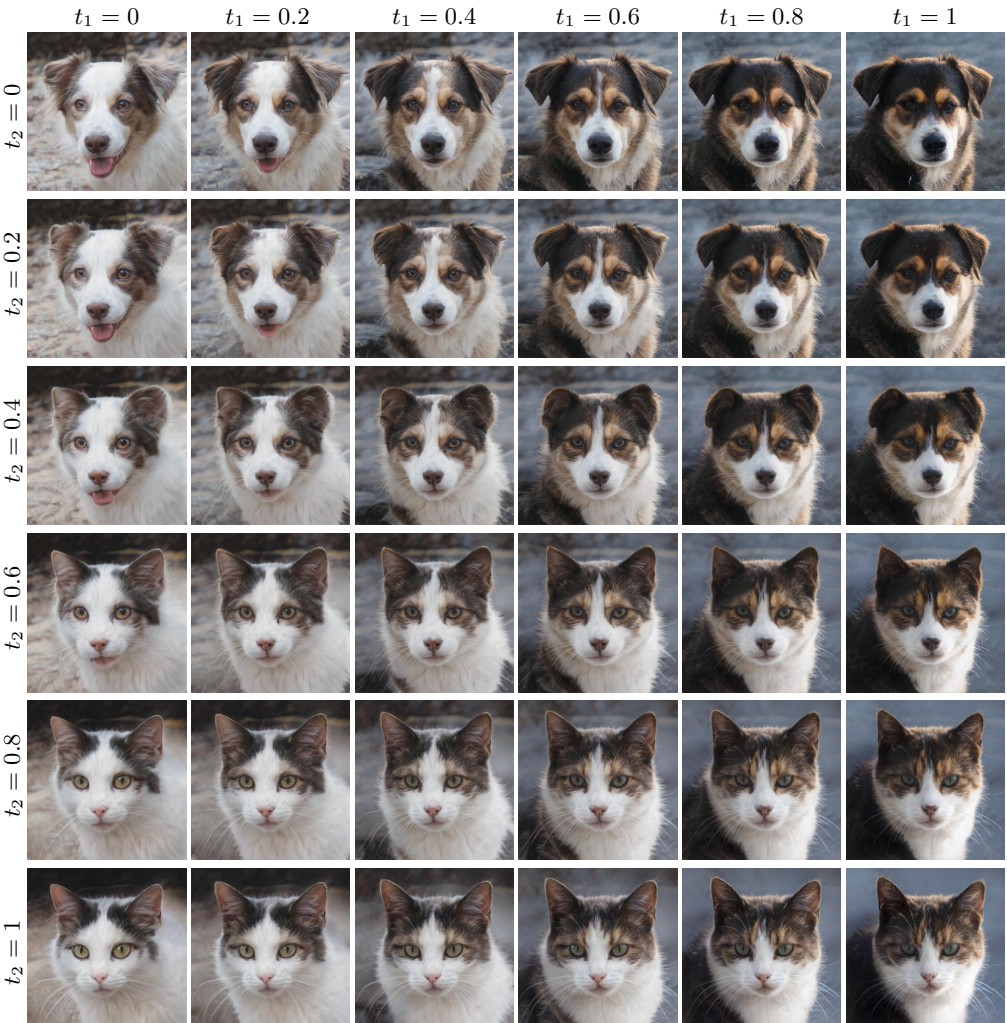

Figure 25: Given a pair of real images from domain $A$ (top-left) and $B$ (bottom-right), we smoothly transition between them by interpolating their latent codes in $\mathcal{W}+$, as well as the model weights. $t_1$ is the interpolation coefficient for the latent codes, while $t_2$ is the coefficient for the model weights. In the same column (fixed $t_1$), we obtain a smooth transition between the domains (different species, but the same pose and fur color). In the same row (fixed $t_2$), we have a smooth transition inside the same domain (same species, varying pose and fur color). Any trajectory between the top-left and bottom-right corners yields a smooth morph sequence between two input images. See the accompanying video, which progresses along the diagonal $t_1 = t_2$.

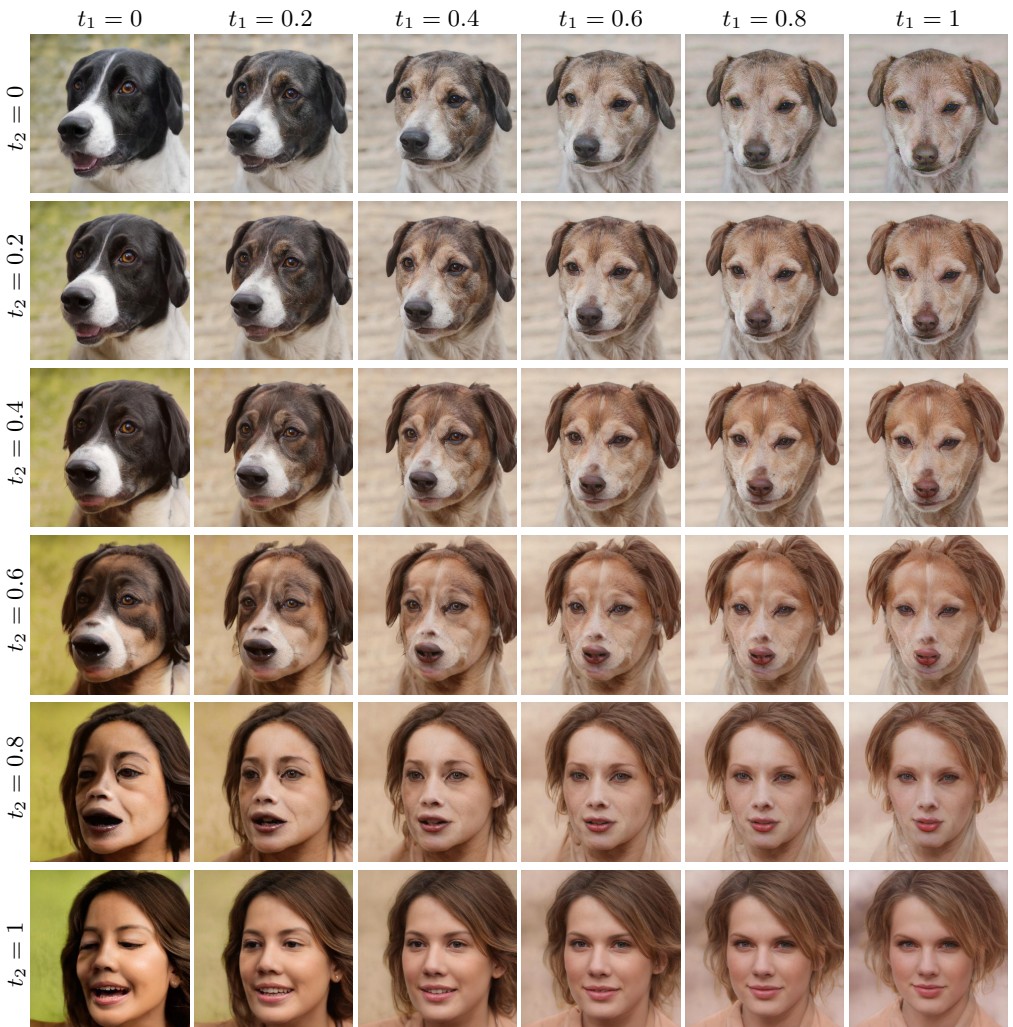

Figure 26: Given a pair of real images from domain $A$ (top-left) and $B$ (bottom-right), we smoothly transition between them by interpolating their latent codes in $\mathcal{W}+$, as well as the model weights. $t_1$ is the interpolation coefficient for the latent codes, while $t_2$ is the coefficient for the model weights. In the same column (fixed $t_1$), we obtain a smooth transition between the domains (different species, but the same pose and similar fur/hair color). In the same row (fixed $t_2$), we have a smooth transition inside the same domain (same species, varying pose and color). Any trajectory between the top-left and bottom-right corners yields a smooth morph sequence between two input images. See the accompanying video, which progresses along the diagonal $t_1 = t_2$.

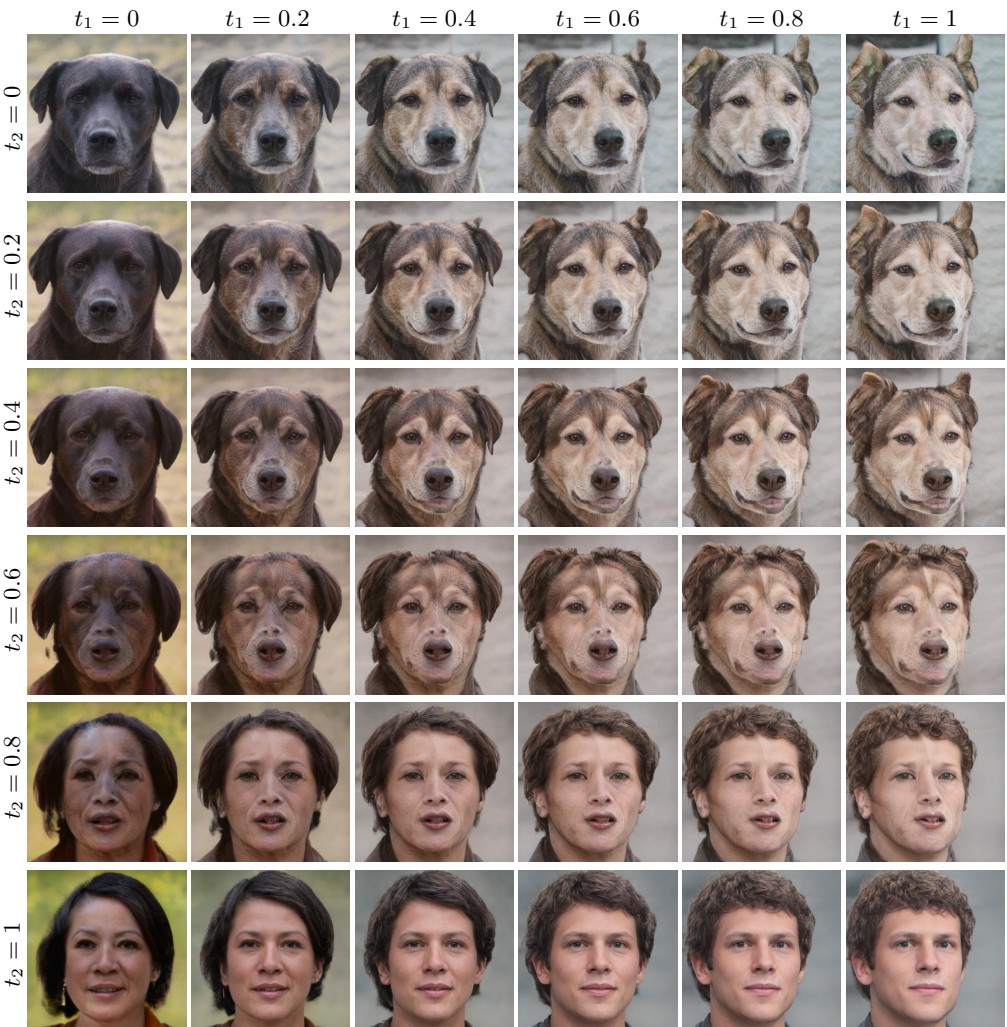

Figure 27: Given a pair of real images from domain $A$ (top-left) and $B$ (bottom-right), we smoothly transition between them by interpolating their latent codes in $\mathcal{W}+$, as well as the model weights. $t_1$ is the interpolation coefficient for the latent codes, while $t_2$ is the coefficient for the model weights. In the same column (fixed $t_1$), we obtain a smooth transition between the domains (different species, but the same pose and similar fur/hair color). In the same row (fixed $t_2$), we have a smooth transition inside the same domain (same species, varying pose and color). Any trajectory between the top-left and bottom-right corners yields a smooth morph sequence between two input images. See the accompanying video, which progresses along the diagonal $t_1 = t_2$.

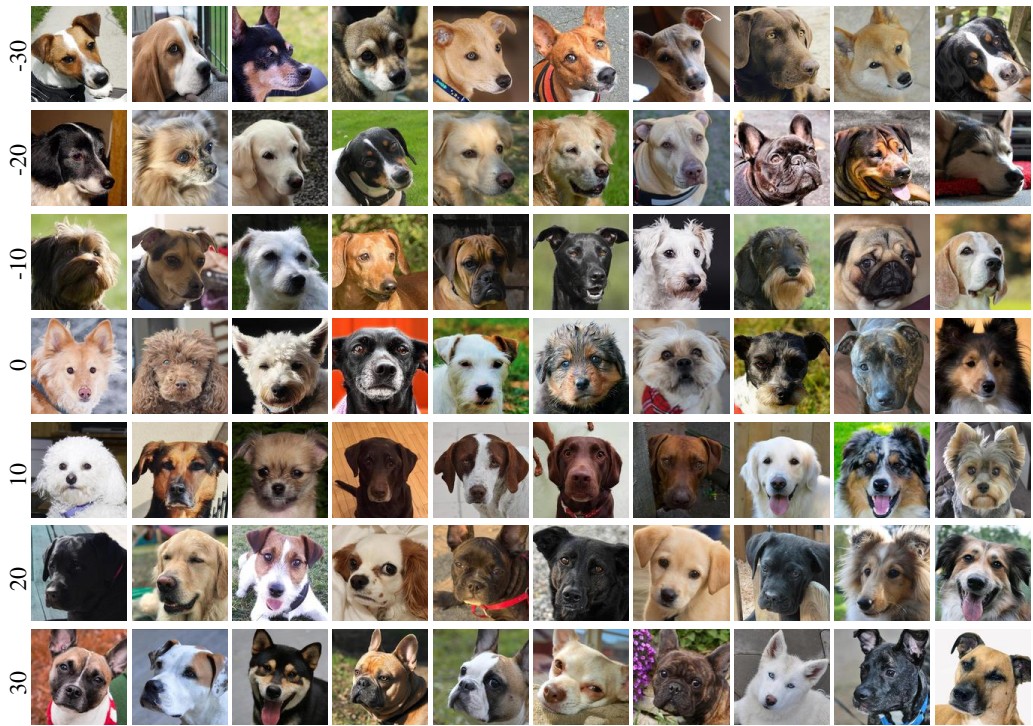

Figure 28: Demonstration of our zero-shot dog yaw regression model. The images are from AFHQ dog dataset, split into several bins (rows), based on the regressed yaw values. The images shown are randomly picked from each bin (no cherry picking). The estimated yaw values capture the correct tendency (right facing to left facing), and in most cases appear to be close to the actual yaw degree.

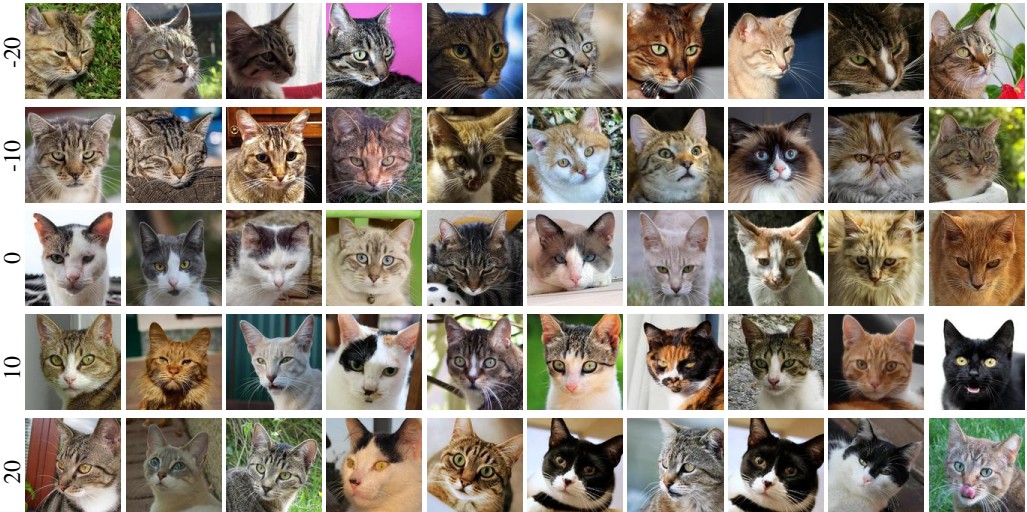

Figure 29: Demonstration of our zero-shot cat yaw regression model. The images are from AFHQ cat dataset, split into several bins (rows), based on the regressed yaw values. The images shown are randomly picked from each bin (no cherry picking). The estimated yaw values capture the correct tendency (right facing to left facing), and in most cases appear to be close to the actual yaw degree.

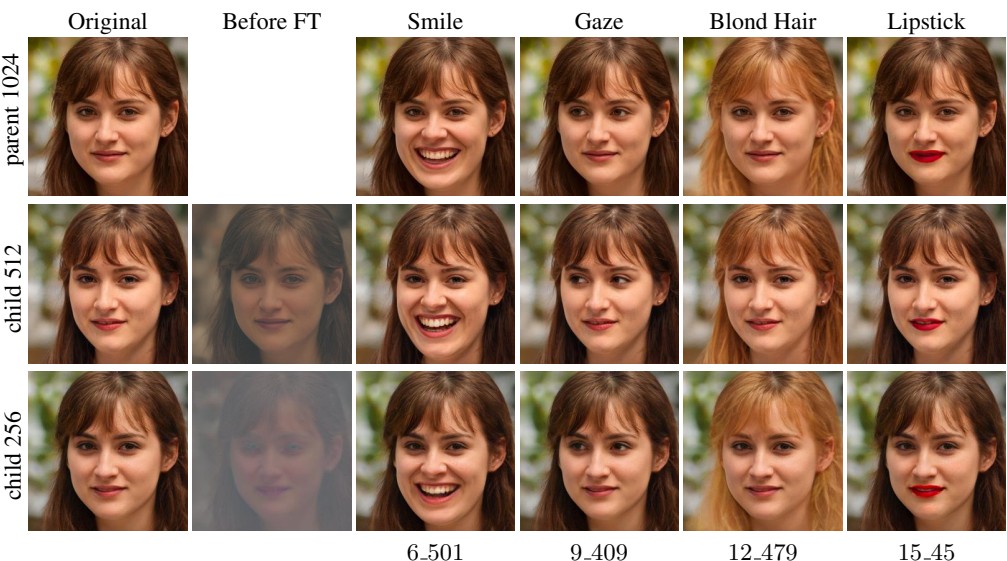

Figure 30: Starting from a pretrained StyleGAN2 model for FFHQ $1024 \times 1024$ resolution as parent, we use its weights to initialize models for $512 \times 512$ or $256 \times 256$ resolution. Before fine tuning (FT), it only generates low contrast images. After fine tuning ("Original" column), similar images with the same attributes (identity, hair length, gender, etc.) as parent model are generated given the same code $z \in \mathcal{Z}$. Note that the generated images are not pixel-wise identical, but the different style channels retain their semantic function, as demonstrated by the four rightmost columns.

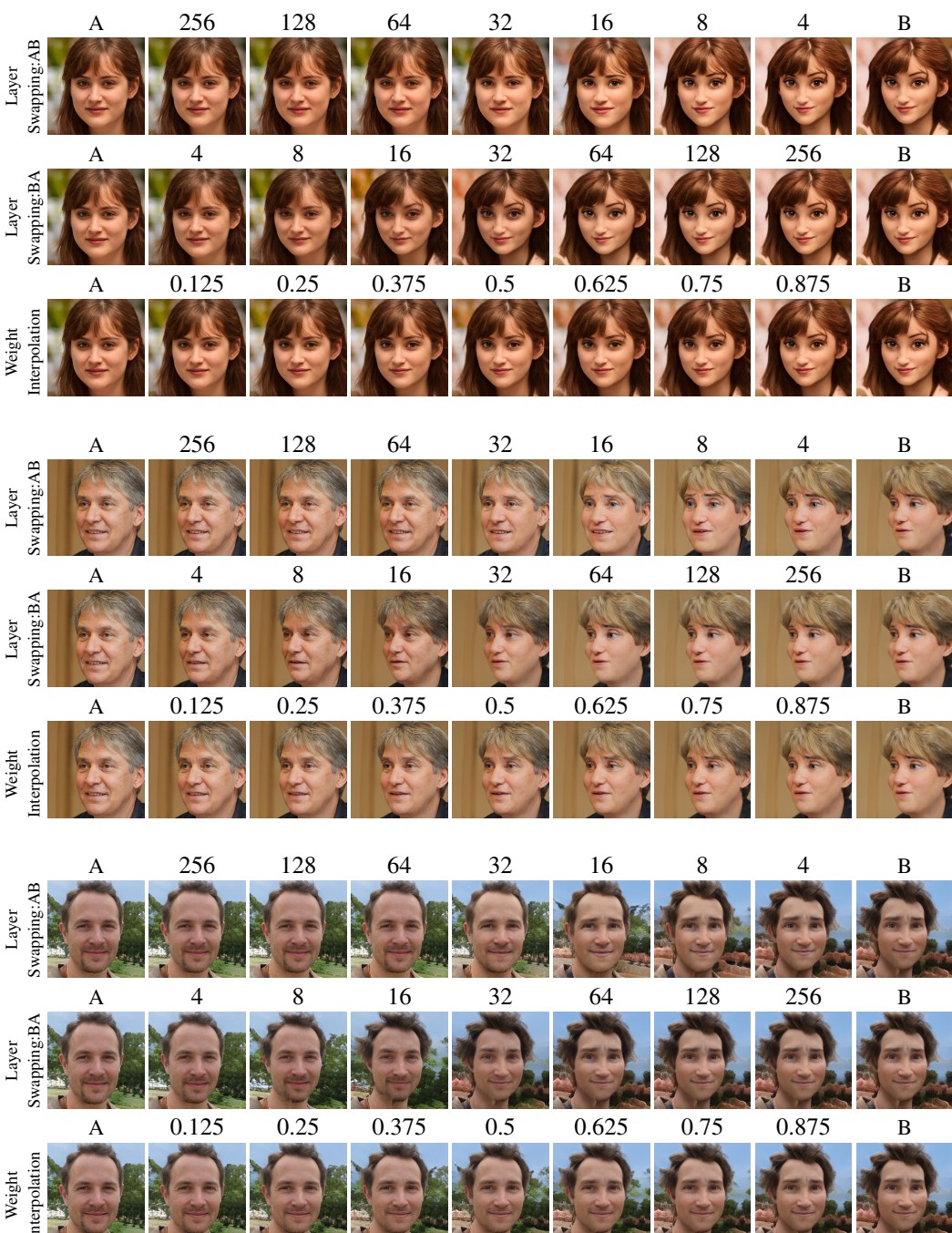

Figure 31: A comparison between layer swapping and model weight interpolation. We demonstrate transitioning between FFHQ and Mega using three different ways. *Layer swapping:AB* means using a hybrid model whose low resolution layers come from model A, and high resolution layers from model B, while *Layer swapping:BA* means the opposite roles (low from B, high from A). The resolution at which the switching occurs is shown above each result. The swapping resolution used by Toonify (Pinkney & Adler, 2020) is either $16 \times 16$ or $32 \times 32$. *Weight interpolation* instead linearly interpolates the weights of all layers between model A and B. The interpolation ratio is shown shown above each result.

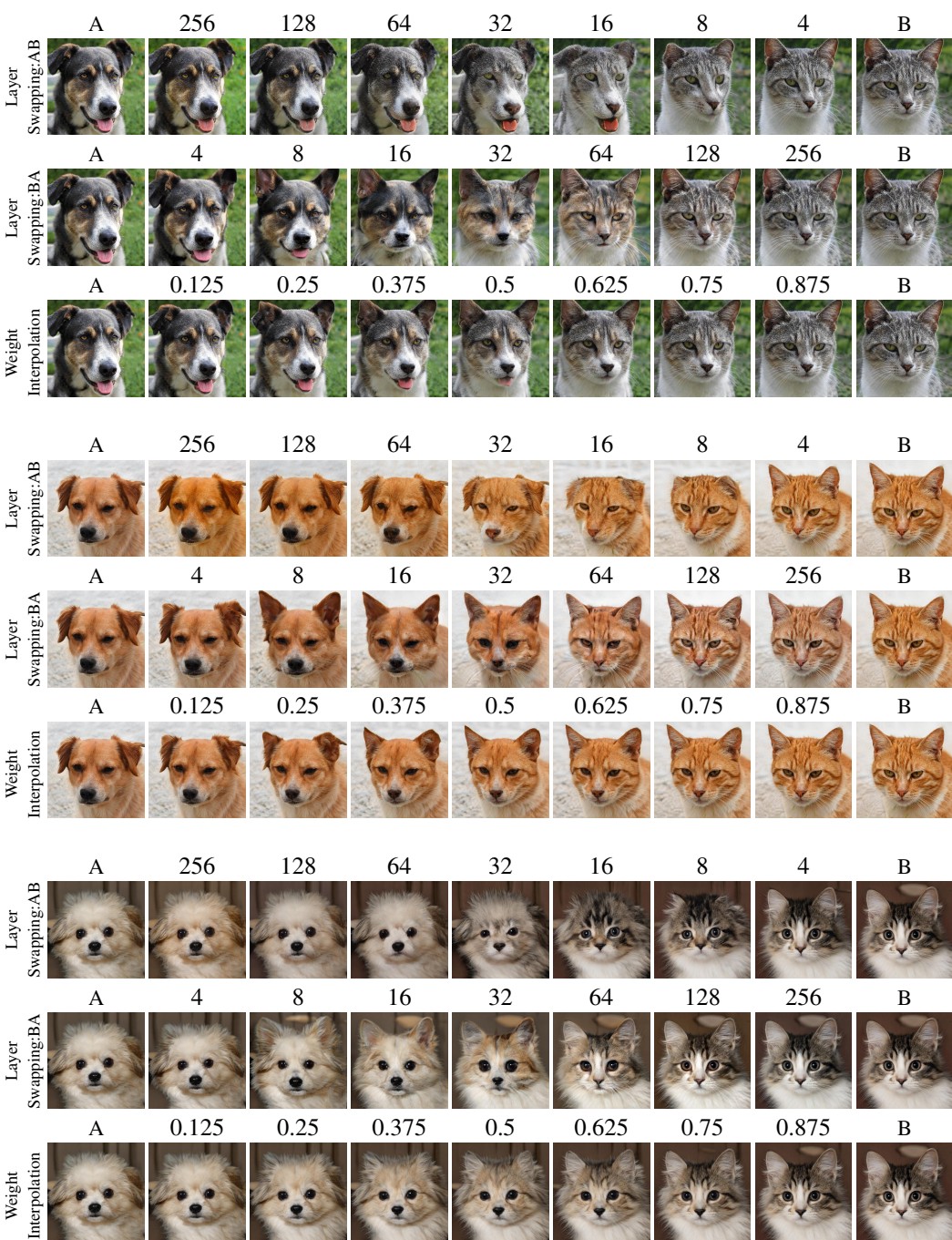

Figure 32: A comparison between layer swapping and model weight interpolation. Here we demonstrate transitioning between AFHQ dog and cat. Refer to Figure 31 for more details.

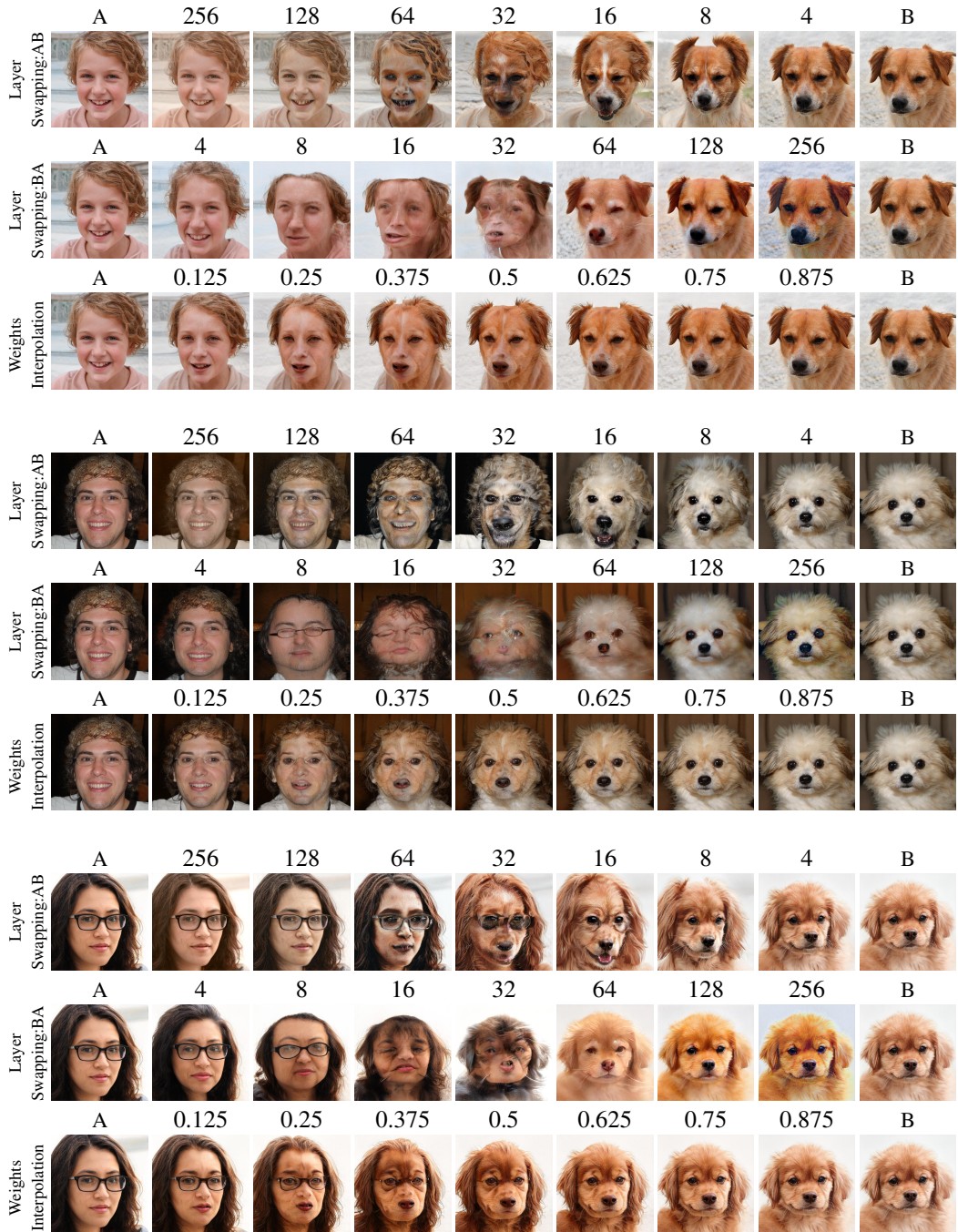

Figure 33: A comparison between layer swapping and model weight interpolation. Here we demonstrate transitioning between FFHQ and AFHQ dog. Refer to Figure 31 for more details.

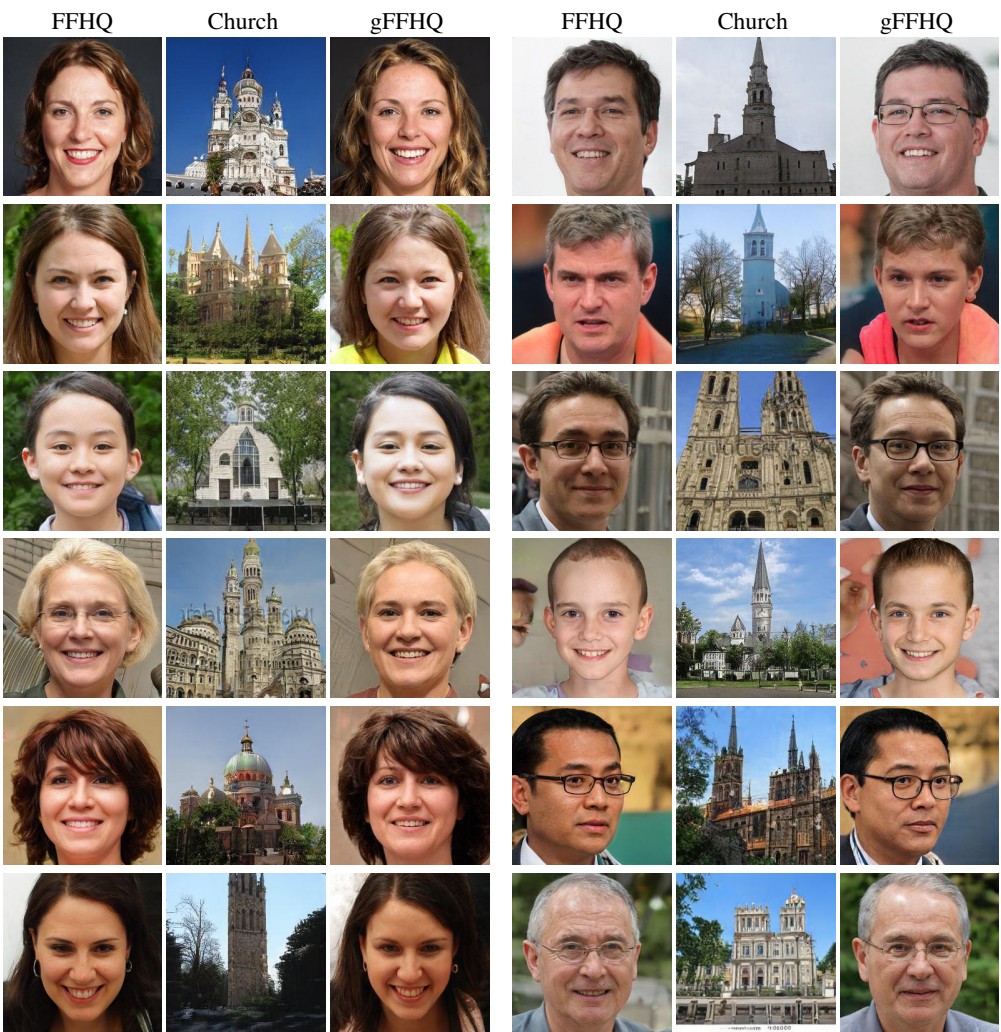

Figure 34: Generative image translation from parent FFHQ model to child LSUN church model and grandchild FFHQ using the same latent code $z$. Since the domain gap between FFHQ and LSUN church is too large, we can barely see any correspondence. But the parent FFHQ model and grandchild FFHQ models generate faces with highly similar attributes and identity. This implies that knowledge that was not transferred from task A (FFHQ generation) to task B (LSUN church generation), is only hidden in the latter model's latent space, rather than forgotten.

Original     Beard     Black Hair

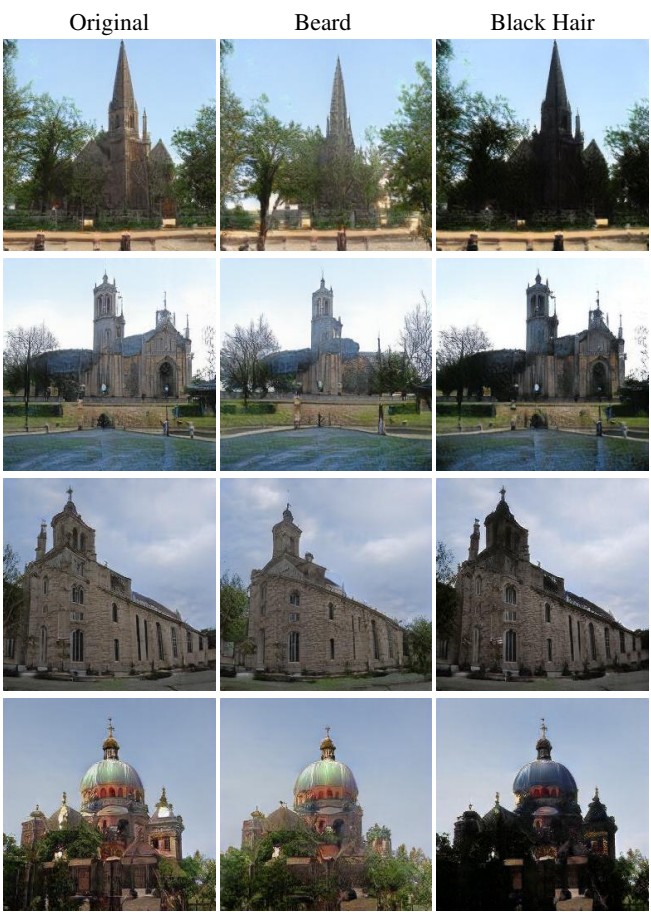

Figure 35: Applying the "Beard" and "Black hair" manipulation directions from parent FFHQ model to a child LSUN church model. The manipulation directions are discovered by StyleCLIP (Patashnik et al., 2021). Most manipulation directions from the FFHQ parent do not change anything in the child church model. Surprisingly, the beard direction from FFHQ appear to control the amount of trees in the church model to some extent, and the black hair direction from FFHQ makes the church building darker in the child model.

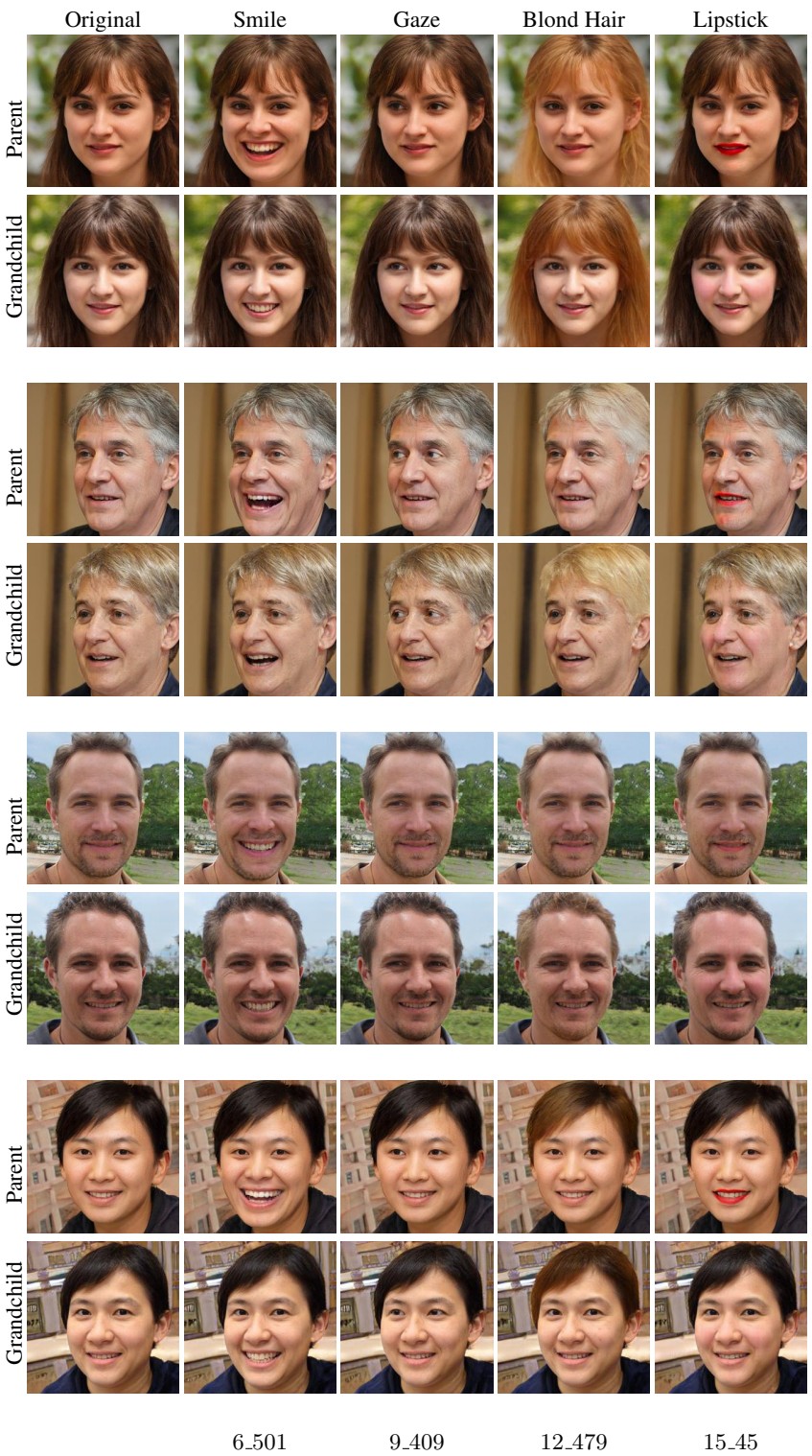

Figure 36: Semantic alignment between parent and grandchild model. We first train a StyleGAN model on FFHQ (parent), then fine tune on LSUN church (child), and finally fine tune back to FFHQ (grandchild). We can see that the same channel still controls the same attribute between parent and grandchild model. Interestingly, channel 15_45 controls lipstick in the parent model, but makes the face slightly pink in the grandchild model. Although the exact function has changed after fine tuning, it is still semantically related to the original function.

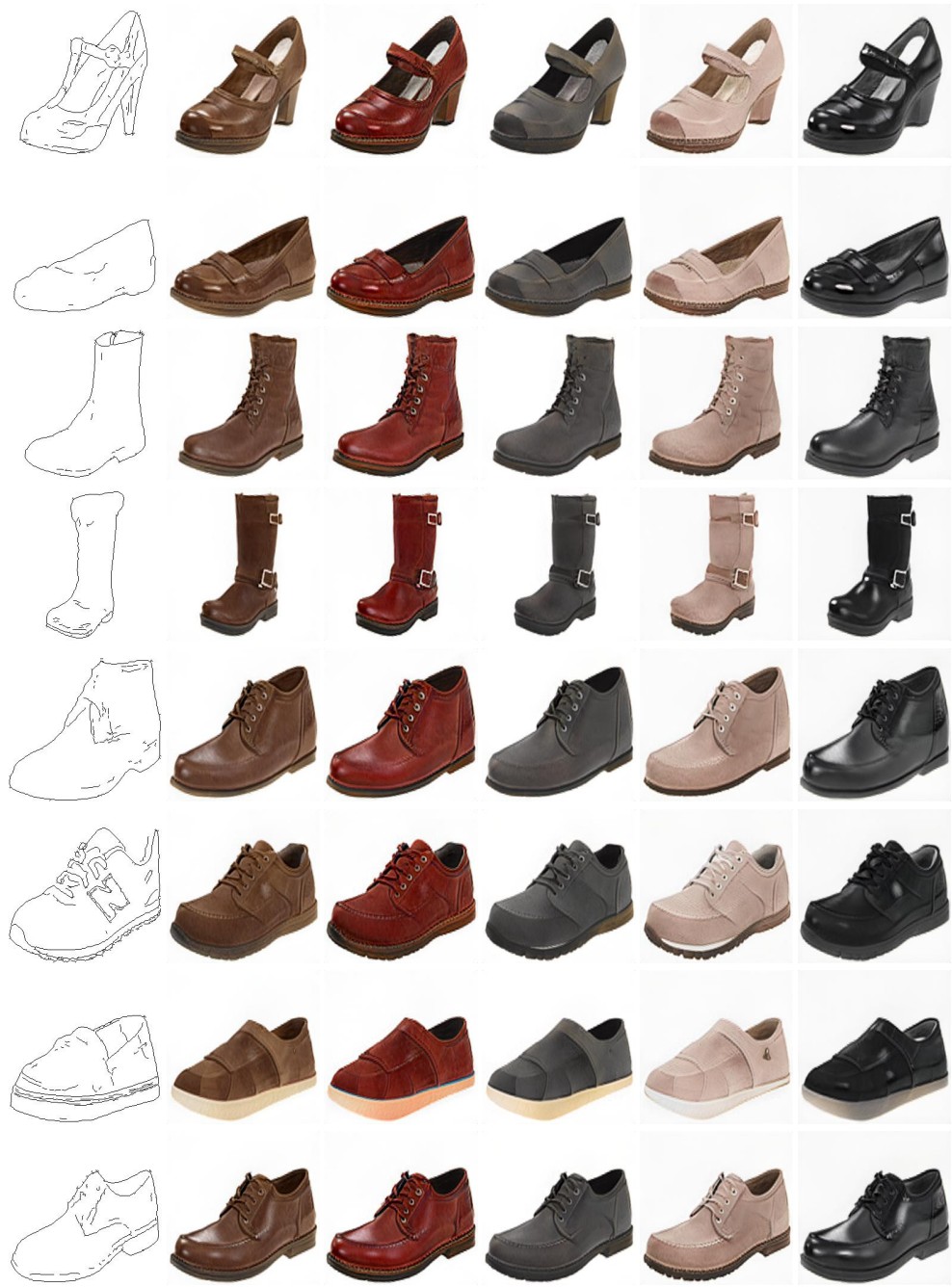

Figure 37: We demonstrate the ability of our method to perform I2I tasks that only change texture, while preserving the structure. Although this dataset has paired edge maps and shoe images, the pairing information is not used by our method. We slightly blur the edge maps to make the images more continuous. We first train a StyleGAN model on the shoes dataset (parent), then fine tune on edge maps dataset (child). Since edge maps mostly represent the structure of objects, and do not contain color or texture, we train an e4e encoder to $\mathcal{W}+$ from whose output we only use the parts that control generator resolutions below $32 \times 32$ (same as was done for multi-modal image translation). The parts that control higher-resolution layers are sampled, yielding multiple possible shoe images (sharing the same structure) for each edge map.

|             | Mega  | Metface | Dog   | Cat   | Wild  | Unrelated FFHQ |
|-------------|-------|---------|-------|-------|-------|----------------|
| L1 in $\mathcal{W}$ | 0.033 | 0.057   | 0.162 | 0.172 | 0.141 | 0.391          |

Table 7: Average L1 distance between $w \in \mathcal{W}$ vectors mapped from the same latent code $z \in \mathcal{Z}$ for different pairs of models. Using a pretrained FFHQ model as parent, it is fine-tuned on different datasets separately. We sample 100K random $z$ vectors and compute the corresponding $w$ for each model. The mean change (per coordinate of $w$) is reported for each child model. It may be clearly seen that in models fine-tuned to nearby domains (Mega, Metface) the change in $w$ is much smaller than to more distant domains (Dog, Cat, Wild), and an order of magnitude smaller than the difference to another FFHQ model, trained independently. These results quantitatively demonstrate that the change in the mapping function is very small for similar domains, larger for more distant domains, but even for distant domains the mapping functions are more closely related than those of two separately trained models.

