# OpenReview forum: "StyleAlign: Analysis and Applications of Aligned StyleGAN Models"
_ICLR.cc/2022/Conference — ICLR 2022 Oral_

### Official Review · Reviewer_nPWN · 2021-10-27

**Correctness:** 4
**Technical Novelty And Significance:** 2
**Empirical Novelty And Significance:** 4
**Recommendation:** 8
**Confidence:** 4

**Main Review:**

Weaknesses

	The main weakness for this paper is likely the limited novelty in the technical contributions, given that transfer learning is a fundamental, established technique. However, this paper is more of a study of this technique, applied to the particular domain of image GANs, and how their editability changes during the process.

	The qualitative results do seem high quality, but it is sometimes quite difficult to see where the resemblance to the reference images are (especially with Figure 6, for the dogs dataset). Perhaps the translation is not focused strongly enough on the reference images. Despite more artefacts being present, I'd say OverLORD images are a better transfer, for both cats and dogs.



Strengths

	The primary contribution of the paper is the extensive experimentation, around transfer learning in GANs, and they perform this very well. Their approach is very well documented, is very thorough, and convincing. The supplementary videos are helpful in further displaying their results.


	Although there are aspects that are both better and worse, the samples from their models compared to other models, are mostly good. The benefits arise mostly from the editability of the output, given their findings that the channels maintain their editing effects from previous datasets with more labels.

**Summary Of The Paper:**

The paper undertakes extensive experimentation in the adaptability of latent space modifications from a base model, and a fine-tuned model for a secondary dataset. They demonstrate which areas of the models change most, and which information remains trained in the parameters, despite re-training. They go on to use their findings in downstream experiments, reaching state-of-the-art quality.

**Summary Of The Review:**

The paper presents some novel analysis in a popular field of research, with implications that can help drive the field further, with minimal changes.

---

> ### Author Response · Authors · 2021-11-20
> **reply to reviewer nPWN**
>
> We thank the reviewer for time and thoughtful feedback. Below we address the main points raised in this review.
>
> * Novelty: indeed, as pointed out in most of the reviews, the main novel contribution of our paper lies in the extensive empirical analysis of transfer learning in the case of the StyleGAN architecture and the applications that become possible due to the semantic alignment of the latent spaces.
>     1. Specifically, we show that multiple channels maintain their function after fine tuning, for both nearby and distant domains. This single channel level analysis is more fine grained than previous works.
>     1. We propose a way to quantitatively measure the channel alignment between parent and child.
>     1. We show the knowledge from the parent model that is not utilized in the child model remains inactive in the latent space, rather than forgotten.
>
> * We agree that OverLORD’s results sometimes resemble the reference image more closely than our results. On the other hand, they don’t always capture well enough the pose or the structure of the source image (for example, the eyes of the source dog are open in the top left example, but in OverLORD’s result the eyes are closed, as in the reference; in the bottom right the ears of the source are erect, but OverLORD’s result takes the shape of the ears from the reference). We believe our method, StarGAN-v2 and OverLord, win in different aspects. This is due to the ambiguity of content-style separation. We believe a more fine-grained version of disentanglement should be (pose, structure, texture, color) separation, where the different aspects are explicitly. A good disentanglement method should allow users to create images based on these 4 explicit aspects (pose, structure, texture, color), rather than the ambiguous (content, style) aspects.
>     1. StarGAN-v2 takes the (pose, structure) from source images, and creates unrealistic structures in the reference domain (1st example in dog2cat, all examples in wild2dog).
>     1. OverLORD takes mainly the pose from source images (for wild2dog case, part of pose is taken from reference), therefore it preserve the appearance of the reference well, but sometimes fail to capture the pose and structure (e.g., ear shape) from the source image (2nd and 3rd examples in wild2dog).
>     1. Our method takes the pose from the source domain, and adapts the source structure to the reference domain (therefore it will not create unrealistic structures in the reference domain), the texture and color are from the reference domain. Our method thus attempts to create a balance between source and reference domain.

---

> > ### Comment · Reviewer_nPWN · 2021-12-02
> > **Thank you for the reply**
> >
> > I maintain my stance that the paper should be accepted - this is a valuable study

---

### Official Review · Reviewer_x6dT · 2021-10-31

**Correctness:** 3
**Technical Novelty And Significance:** 2
**Empirical Novelty And Significance:** 3
**Recommendation:** 6
**Confidence:** 5

**Main Review:**

Pros:
1. The shared latent space for "child" and "parent" networks is interesting, and matches the learning representation goal.

2. The overall idea and the proposed fine-tuning method towards analyzing the shared latent space is reasonable. The StyleGAN-based architecture is a powerful structure, which consists of a mapping network, an affine translation, and a generator. This is exactly suitable for exploring feature attributes at different levels.

3. The paper provides many interesting visual results, such as in Figures 1, 2, 3, and 4, to show the effectiveness of shared information in different networks. These interesting conclusions may contribute to the generation community to design better generator works.

4. The wide applications are being met by subtly using the shared semantic information between different domains.

Cons:

While the impressive results are achieved, I believe some parts need more clarification (even after considering the supplementary material):

1. The key limitation of this work is the novelty. If I understand correctly, the authors just fine-tuning the different parts of the StyleGAN. This is very similar to the existing works AgileGAN and the shared attributes have also been mentioned in prior MUNIT. It takes me a hard time to buy the novelty of models, techniques, and theoretical insights.

2. The authors claimed the "two W spaces are point-wise aligned" on page 4. While they showed some shared attributes in Figure 2 and Table 1, and they also demonstrated some attributes are hidden, I do not believe these attributes are "point-wise" aligned. Is this conclusion correct? How to demonstrate it?

3. A quantitative and qualitative comparison with the latest CUT and F-LSeSim is given in Figure 5. While the improved results are provided, the original CUT and F-LSeSim work for aligned shapes. Why not provide some aligned examples, for example, horse2zebra, night2day, apple2orange, every two domains share a similar shape, but different appearances? This would be more robust to demonstrate the effectiveness of aligned style on both aligned examples (Figure 5) and unaligned examples (Figure 6).

4. In Figure 6, the StarGAN2 seems to be able to provide more consistent results to the source image, such as the smiling mouth in the first row, and the same head poses for the third row. Except for the FID score, it seems the StratGAN2 provides better translated results (better shared content, such as pose and expression).

5. The authors show abundant results in Figures 23, 24, 25, and 26. Similar results are also being met in the StyleGAN-based method. I do not fully understand the key challenge in such a situation.

**Summary Of The Paper:**

This work is about the task of transfer learning to tame a new "child" network using a pre-trained "parent" network. While the model and fine-tuning technology lack novelty, the shared semantic information in the generation network are interesting. Finally, the authors applied the proposed aligned model to multiple tasks, including image-to-image translation, cross-domain image morphing, and zero-shot classification and regression. The impressive results with shared semantic information are achieved.

**Summary Of The Review:**

As can be inferred from the balance in strengths/weaknesses, my preliminary rating for this submission is borderline accept. While the interesting results are shown in the paper and supplementary material, the key contribution for models, techniques or theoretical insights is unclear. Furthermore, some comparisons seem unfair in the paper. The authors should clearly interpret them during the rebuttal.

---

> ### Author Response · Authors · 2021-11-20
> **Reply to reviewer x6dT**
>
> We thank the reviewer for time and thoughtful feedback. Below we address the main points raised in this review.
>
> 1. Indeed, our work focuses on simply fine-tuning StyleGAN. Nevertheless, as pointed out in most of the reviews, our main novel contribution comes from the extensive empirical analysis of aligned models and their use for downstream tasks. As we discuss in the paper at length, several previous works [1,2,3], including AgileGAN [4], have used “fine-tuning” for image-to-image translation. Nevertheless, we go further than that: we analyze several aspects of aligned models and discover novel insights regarding their nature. We further apply aligned models for additional tasks and demonstrate good performance.
> We also agree that MUNIT [5], as well as others [6,7], have discussed shared attributes in latent space for image-to-image translation (we mention it in second paragraph on page 4). However, we note that such works have devised a dedicated architecture to encourage a shared latent space. Conversely, in our framework this phenomenon occurs without any task-specific encouragement or intervention.
>
> 2. We observed that the mapping network changes very little during fine-tuning. This implies that a single z code is mapped to similar w codes. While the correspondence of images is demonstrated in the paper (Figure 1), we agree that the similarity of w codes is not explicitly shown. According to your feedback, we have now performed an additional experiment in order to show this. We measure the L1 distance between w codes inferred by different mapping networks from the same z code. The results (reported in a new Table 7, end of appendix) clearly demonstrate that indeed the w codes are very close in latent space for models that are closely related. We refer to this as “point-wise alignment”, but indeed perhaps a better term could be used - we’re open to suggestions.
>
> 3. We consider working *only* with aligned shapes a limitation of several existing I2I methods. We thus believe comparison on domains with different shapes to be crucial. Nevertheless, we agree a comparison on aligned shapes would improve our evaluation more complete. We used our approach to translate edge maps to shoe images (see new Figure 37). We note that we do not rely on the paired information (like pix2pix and similar).
> It may be seen that our method maintains the structure well, and the generated texture is realistic. Due to time constraints, we were not able to train CUT or F-LSeSim yet, but will include the comparison in a future revision.
>
> 4. As previously mentioned, a known limitation [8] of several I2I methods is that they are unable to change shape and instead change mostly the texture. This indeed causes StarGAN-v2’s results to be more similar to the source image. However, this comes as the cost of generating images that are less realistic for the target domain. For example, in Figure 6, on the second and third row on the right, the generated images have a structure of a wild animal and not of a dog, detracting from their realism. Unfortunately, the task of image-to-image is ill-defined in the first place. We believe our method, StarGAN-v2 and OverLord, each achieve different aspects. Please see our response to reviewer nPWN, where we discuss this in more detail.
>
> 5. Figures 24-27 demonstrate cross-domain image morphing. StyleGAN inherently supports in-domain image morphing through interpolation of latent codes. However, a single StyleGAN can only generate images in a limited domain. Training a single StyleGAN on both FFHQ and AFHQ dogs will not result in high quality generated images. Therefore, traditional StyleGAN-based methods can morph well between human faces or between dog faces, but not between humans and dogs. In contrast, we perform cross-domain image morphing (human face and dog face) using aligned models. Specifically, we interpolate the model's weights and latent codes simultaneously. We show a smooth transition from dog2cat in Figure 24 and 25, dog2human in Figure 26 and 27.
>
>
> [1] Pinkney/Adler. "Resolution Dependent GAN Interpolation for Controllable Image Synthesis Between Domains."
>
> [2] Kwong et al. "Unsupervised Image-to-Image Translation via Pre-trained StyleGAN2 Network."
>
> [3] Wang et al. "Deep network interpolation for continuous imagery effect transition."
>
> [4] Song et al. "AgileGAN: stylizing portraits by inversion-consistent transfer learning."
>
> [5] Huang et al. "Multimodal unsupervised image-to-image translation."
>
> [6] Liu et al. "Few-shot unsupervised image-to-image translation."
>
> [7] Liu et al. "Unsupervised image-to-image translation networks."
>
> [8] Gabbay/Hoshen. "Scaling-up Disentanglement for Image Translation."

---

> > ### Comment · Reviewer_x6dT · 2021-11-29
> > **Thanks for the authors' reply**
> >
> > Thanks for your reply. All my concerns have been addressed. While the fine-tuning method is the same as prior works, I agree with other viewers that the analysis of the latent space is interesting and can contribute to other related researches.

---

### Official Review · Reviewer_fnkZ · 2021-10-31

**Correctness:** 3
**Technical Novelty And Significance:** 3
**Empirical Novelty And Significance:** 3
**Recommendation:** 8
**Confidence:** 4

**Main Review:**

**Strengths:**
+ There are many works using aligned StyleGAN models but without deep analyses on the properties. This paper provides an in-depth study of the properties, which provides some insights on the aligned models and might inspire following more complex researches.
+ The organization of the paper is good and the analyses are comprehensive. Several observations are studied with quantitative/visual analyses to prove key properties, followed by corresponding applications utilizing the properties, making the paper easy to follow and concrete.

**Weaknesses:**
+ While the part of the property study is comprehensive and interesting and provides insights, the part of the application is less insightful. The application involves the image translation, image morphing and zero-shot vision tasks.
  - Image translation and image morphing are intuitive and well-studied in previous studies, as also pointed out in the paper, which might provide limited insights. Image morphing by interpolating aligned models has been studies in `[R1]`.
  - The zero-shot vision tasks involve utilizing the label of the parent domain to edit/classify the child domain, which gives some novel ideas but alone are not very thorough to me.
  - I think it will be valuable to investigate the applications in terms of the specific properties. For example, only the conv parts of network change greatly for close domains, which supports translations. Can we just finetune the conv part and fix all other parts for better translations? (which is also discussed in `[R3]`) The paper discusses that the latent semantics are hidden rather than forgotten during finetune. It will be valuable to also discuss the potential application of this property.

**Some small issues:**
+ In Fig. 14, the authors demonstrate the semantic alignment between the human face and the church. However, only the church images are shown. The corresponding face images using the same latent z are suggested to be added to help the readers better find their correspondences.
+ In Sec. 4.2, the paper claims that the layer swapping performs the transition as a series of discrete steps, rather than continuously. In the implementation of Pinkney `[R2]`, the layers can be smoothly swapped through the parameter ` blend_width` instead of hard swap, which might be able to perform the transition continuously.

`[R1] 2019 CVPR Deep Network Interpolation for Continuous Imagery Effect Transition`

`[R2] https://github.com/justinpinkney/toonify/blob/master/StyleGAN-blending-example.ipynb`

`[R3] 2021 TMM Unsupervised Image-to-Image Translation via Pre-trained StyleGAN2 Network`


**Summary Of The Paper:**

This paper provides an in-depth study of the properties and applications of the semantic alignment between the original parent StyleGAN model and its finetuned child model on another dataset. Specifically, the paper empirically demonstrates the semantical alignment of the two models. Then, based on the properties, the paper solves serval tasks like image translation, image morphing, zero-shot image editing and attribute classification.

**Summary Of The Review:**

This paper investigates an important topic of the semantic alignment of the parent model and finetuned child model. Despite the application part could be further enriched to better match the analyzed properties, the comprehensive analyses on the properties provides many insights and might arouse following researches. Therefore, I am positive.

---

> ### Author Response · Authors · 2021-11-20
> **reply to reviewer fnkZ**
>
> We thank the reviewer for time and thoughtful feedback. Below we address the main points raised in this review.
>
> * While the focus of our work is on the extensive empirical analysis of transfer learning in the case of the StyleGAN architecture and the applications that become enabled by semantic alignment of latent spaces, we do believe that there are also several contributions in the applications themselves:
>     1. We show the Toonify based method can work for more distant domains (dog2cat), rather than just nearby domains (ffhq2cartoon).
>     1. The Toonify based methods are mainly used in I2I tasks, we show its ability for multi-modal image translation tasks.
>     1. In supplementary A.4 and Figure 30, we propose a way to obtain a low-resolution child model from a high-resolution parent model, while keeping the semantics of each channel. This saves considerable computation time, compared to training and analyzing an unrelated low-resolution model.
>
> * Thanks for referring us to [R1]. We were not familiar with this work at the time of writing and have now revised the text to cite it properly (See page 8). In addition to the revised text, we want to explicitly point out that [R1] discusses a transition between the effects of two different networks. For example, the neworks might do style-transfer for different styles, and their method is able to create styles in between. We, on the other hand, morph two real images, possibly from distant domains. For example, creating a hybrid between a person and a dog. For this end, beyond weights interpolation we also invert the real images and interpolate the latent codes.
>
>  * Although [R1] shows the filter patterns of the same channel remain similar after fine tuning, we show the actual function of the channel remains the same. We believe that examining the channel function is more important than its filter pattern, since two channels may have the same filter pattern but completely different function.
> [R1] also shows weight interpolation between aligned models, but mainly for nearby domains (day2night, image style transfer), which only requires changes in color and texture.  We show results for distant domains (dog2cat, ffhq2dog), which require changes in both structure and texture. It is worth mentioning that we interpolate model weights and latent codes simultaneously, which allows us to interpolate two unrelated images (a black cat and a yellow dog with different pose).
>
> * Can we just finetune the conv part and fix all other parts for better translations?
> This essentially would result in a StyleGAN model custom-tailored to the translation task. We did a related experiment, but the performance was unsatisfactory. Specifically, when using aligned models to do real image translation, there are two issues:
>     1. The latent space (W+) of the parent and child model may not be well aligned, since the mapping function may change during fine tuning. Inverting the real image into Z space may alleviate this problem, but at the cost of reconstruction quality (the translated image might not resemble input image well enough).
>     1. The generative W+ distribution may not be the same as the W+ distribution of inverted images. For example,  [1] show that latent optimization in W+ space results in latent codes that lie outside the generative distribution. Manipulating these latent codes creates significant artifacts in image space.
> We have attempted to resolve these issues for cartoonization, but the resulting model suffered from overfitting the noise present in the child domain (Mega). We leave further experiments to future research.
>
> * Application of “the hidden latent semantics” between parent, child and grandchild
> We believe this property could be used as a “watermark” for a trained model. If someone uses without permission a trained model (parent) to train another model (child) on some other dataset, it can be shown that the child model is inherited from that parent model through retraining it to the parent’s dataset (grandchild) and measuring the alignment.
>
> * In Figure 34, we add corresponding images between FFHQ parent LSUN church child. More results could be found in Figure 35 and 36.
>
> * We were not aware of the ‘blend_width’ parameter in Pinkney’s implementation, thank you for pointing it out to us! In the Toonify paper (https://arxiv.org/pdf/2010.05334.pdf), equation 2, the interpolation parameter is defined as either 0 or 1, which results in a discrete layer swap, so there seems to be a mismatch between the paper and the implementation.
>
> [1].  Wu, Zongze, Dani Lischinski, and Eli Shechtman. "Stylespace analysis: Disentangled controls for stylegan image generation." Proceedings of the IEEE/CVF Conference on Computer Vision and Pattern Recognition. 2021.

---

> > ### Comment · Reviewer_fnkZ · 2021-11-21
> > **Thanks for the author's reply**
> >
> > Thanks for the author's reply. Part of my concerns are solved. However, I still feel that the applications of image translation and image morphing are less insightful to me.
> >
> > - [R3] has shown the Toonify based method can work for more distant domains (dog2cat).
> >
> > - [R3] has shown the Toonify based method can work for multi-modal image translation tasks.
> >
> > - The authors clarify the paper’s difference from [R1] in terms of implementation details, applicable domains, and channel analysis. From my perspective, I would like to see something new in the applications, which could provide me with some insights. However, since I’m familiar with StyleGAN, Toonify, [R1][R2][R3], I think image-to-image translation and image morphing in this paper are not new to me. For example, it is true that this paper interpolate both model weights and latent codes, while [R1] only interpolates model and StyleGAN only interpolate codes. But that is still not new to me in terms of research values. By comparison, I think obtaining a low-resolution child model from a high-resolution parent model is new to me. (And the property study part is fairly new to me, which I really appreciate.)
> >
> > - [R3] has shown that `Freezing the FC layers (freeze-FC) preserves the mapping relation from a latent code to an input feature vector. Therefore, some semantic information like age and facial expression can be preserved`. The claim is very different from the experiment in authors’ response. Maybe the authors could analyze the reasons why freeze-FC works or does not work to provide more insights.
> >
> > - Thank you for providing this strange application of illegal use detection. I think Figure 34 is very interesting. The grandchild model recovers most of the feature from the parent model, which might have some potential good applications.
> >
> > `[R1] 2019 CVPR Deep Network Interpolation for Continuous Imagery Effect Transition`
> >
> > `[R2] https://github.com/justinpinkney/toonify/blob/master/StyleGAN-blending-example.ipynb`
> >
> > `[R3] 2021 TMM Unsupervised Image-to-Image Translation via Pre-trained StyleGAN2 Network`

---

> > > ### Author Response · Authors · 2021-11-23
> > > **Insights regarding image translation**
> > >
> > > Thanks for responding to our rebuttal.
> > >
> > > * As we acknowledge several times throughout our paper, our I2I workflow is indeed similar to several recent methods [1,2,3]. Our main novel contributions for I2I and multi-modal image translation tasks lie in the analysis of different inversion methods and spaces and the insights resulting from it. Specifically, we decompose the workflow to real image inversion and image translation, and compare multiple alternatives for each of these steps in order to decide which latent spaces should be used in different scenarios. Previous I2I methods [1,2,3] did not provide extensive evaluation of the different components in the process and therefore perhaps unsurprisingly chose different inversion methods and latent spaces. The original Toonify paper [1] only translated generated images (though their github implementation uses direct optimization in W+), [3] uses a different latent optimization objective in W+, while AgileGAN [2] trains an hVAE to Z+ space. In contrast, we perform an extensive evaluation regarding the choice of the latent space and the inversion method, which is briefly summarized below.
> > >
> > >     Specifically,
> > >     1. In Figures 16 and 17, we show that the W+ space, which is good for reconstruction, does not necessarily lead to good translations (wrong color palette). This was discussed by AgileGAN but was not demonstrated. Additionally, we provide a possible explanation - the change in mapping function (shown in Figures 1 and 8). For nearby domains (FFHQ2Cartoon), the mapping function only changes slightly, and the translation results for W+ and Z are compatible (shown in Figure 18 and 19). For more distant domains (AFHQ), the mapping function changes a lot, and the translation results for Z space are significantly better than W+ space (shown in Figures 16 and 17).
> > >     2. Figures 17 and 19 show that inversions using an encoder and using latent optimization achieve images with similar attributes, with latent optimization results having more diversity and better visual quality, as can be seen from the FID and KID results reported in Table 5. This may be because the encoder method effectively compromises over a large set of training images, at the price of losing some image specific features.
> > >
> > > * Regarding whether freezing the mapping (Freeze-FC) works or not, this is not a question that can be easily answered. The difficulty comes from the fact that the current I2I task is not fully defined. Rather than translating the input image from source domain to target domain, users typically wish to translate the source image to a domain that is **between** source and target. For example, the Metface dataset barely contains smiling faces. If we wish to translate an image from FFHQ to Metface, it is reasonable to obtain a non-smiling result. However, this is probably not what the user would consider a satisfactory result.
> > > To create images that mix between source and target domains, Toonify-like methods limit the capacity of the child network through freezing the mapping function [3], or create a hybrid model between parent and child through layer swapping [1,3]. The task is not well defined in the sense that we do not explicitly define which attributes should be maintained and which attributes should be translated.
> > >
> > >     We believe the next generation of image translation methods should allow users to more explicitly choose what to translate and what to maintain. Once the goal of image translation is explicitly defined, we can test and compare the performance of different modifications of the network, such as freezing the mapping function and layer swapping.
> > >
> > > **References:**
> > >
> > > [1] Pinkney, Justin NM, and Doron Adler. "Resolution Dependent GAN Interpolation for Controllable Image Synthesis Between Domains." arXiv preprint arXiv:2010.05334 (2020).
> > >
> > > [2] Song, Guoxian, et al. "AgileGAN: stylizing portraits by inversion-consistent transfer learning." ACM Transactions on Graphics (TOG) 40.4 (2021): 1-13.
> > >
> > > [3] Jialu Huang, Jing Liao, Sam Kwong. "Unsupervised Image-to-Image Translation via Pre-trained StyleGAN2 Network." IEEE Transactions on Multimedia (2021).

---

> > > > ### Comment · Reviewer_fnkZ · 2021-11-23
> > > > **Thanks for the authors' reply**
> > > >
> > > > - I agree with the authors that this paper provides new insights for the choice of inversion methods and spaces for image-to-image translation. However, this content is mainly included in the APPENDIX part, while Sec. 4 mainly describes a comparison with I2I translation methods and image morphing (which is not very new to me), making the readers easily miss the new part. I would suggest the authors to refine and even rephrase Introduction and Sec. 4 accordingly. For example, `Next, we explore additional tasks, for which aligned models have not been used before. In Section 4.2 we describe a simple method for fully automatic image morphing between fairly dissimilar domains, such as human to dog faces` in the introduction is not very precise and is not the key contribution of this paper.
> > > >
> > > > - It seems my concern that the application of image morphing is less insightful is not responded. If the authors agree with my concern, maybe the introduction and experiment of this part could be moderately reduced to leave more room to those really exciting parts of this paper.
> > > >
> > > > - `This was discussed by AgileGAN but was not demonstrated`. AgileGAN verifies that Z+ is better than W+ on face to cartoon translation task by extensive experiments of ablation studies and visualization of the latent code distributions. Why it is not demonstrated?

---

> > > > > ### Author Response · Authors · 2021-11-28
> > > > > **Image morphing, AgileGAN, and paper structure**
> > > > >
> > > > > Thanks for these comments and suggestions,
> > > > >
> > > > > - We do consider it insightful to demonstrate that aligned models may be readily applied to **cross-domain morphing**. Previous (fairly recent) works tackling this task [2,3] resorted to sophisticated machinery for coping with such morphs. As discussed earlier, [1] is related yet not directly comparable, because it does not really **morph** between two real images, but rather interpolates the “effects” of two neural networks. To the best of our knowledge, our method is the first to demonstrate cross-domain image morphing with significant changes in geometry and pose, without explicitly accounting for these factors. We consider the fact that a traditionally challenging task can be solved with such ease, insightful. Furthermore, we also observe at the end of section 4.2 (and demonstrate in Figures 24-27) that there are multiple feasible morphing trajectories in the 2D interpolation space. A creative artist may choose different trajectories depending on the desired transition effect.
> > > > >
> > > > > - AgileGAN discusses that Z+ is superior to W+ for face-to-cartoon translation. Although we arrive at the same conclusion, this claim was not fully supported in the AgileGAN paper, since all of their experiments comparing Z+ to W+ had other varying factors besides the latent space. Namely, to represent W+, they used other inversion methods (pSp and Image2StyleGAN) - differing significantly from their hVAE multi-path encoder. These methods differ in optimization strategy, architecture, losses, etc. These differences make it difficult to attribute the superiority entirely to the choice of latent space.
> > > > >
> > > > > - We understand your point that in regards to image-to-image translation, the analysis part may be more insightful than the comparison to other methods. We consider both important, and ideally would have liked to include both in the main paper, but haven’t been able to do so due to space constraints. We are open to suggestions regarding what should be placed in the paper and what in the appendix, assuming the reviewers are in agreement on this and space constraints permit.
> > > > >
> > > > > [1] 2019 CVPR Deep Network Interpolation for Continuous Imagery Effect Transition
> > > > >
> > > > > [2] K. Aberman, J. Liao, M. Shi, D. Lischinski, B. Chen, and D. Cohen-Or. Neural best-buddies:  Sparse cross-domain correspondence. SIGGRAPH 2018.
> > > > >
> > > > > [3] N. Fish, R. Zhang, L. Perry, D. Cohen-Or, E. Shechtman, and C. Barnes.   Image morphing with perceptual constraints and STN alignment. Computer Graphics Forum 2020.

---

> > > > > > ### Comment · Reviewer_fnkZ · 2021-11-29
> > > > > > **Thanks for the authors' reply**
> > > > > >
> > > > > > - As I pointed out in my previous reply, the subtle detailed differences are not the key factors to me. I think given that [R1] shows aligned model intepolation achieves continuous cross-domain (different styles, different gender) translation, and that [R3] shows layer swapping (a kind of hard interpolation) can adjust the degree of the shape deformation between cats and dogs and the well known StyleGAN latent interpolation to achieve continuous changes in geometry and pose, I do not think the cross-domain morphing in this paper brings me much insights in terms of research values. As also pointed out by Reviewer x6dT, `The authors show abundant results in Figures 23, 24, 25, and 26. Similar results are also being met in the StyleGAN-based method. I do not fully understand the key challenge in such a situation.` It maybe a challenge for traditional image morphing frameworks, but given the StyleGAN framework and related works as well as [R1], the solution to this task is intuitive and simple.
> > > > > > - What I mean in my previous reply is that your claim of `This was discussed by AgileGAN but was not demonstrated` is too absolute. Although not supported in a perfect way, is it fair to claim that AgileGAN does not demonstrate the conclusion (and this is even one of the contributions claimed by AgileGAN) with its analysis, visual comparison and comparison of the latent code distributions?

---

### Official Review · Reviewer_2pb8 · 2021-11-02

**Correctness:** 4
**Technical Novelty And Significance:** 4
**Empirical Novelty And Significance:** 4
**Recommendation:** 8
**Confidence:** 4

**Main Review:**

##########################################################################

Pros:
1. The paper performs the first detailed exploration of model alignment based on StyleGANs. Considering the wide-range applications of StyleGAN, I think the analysis is timely, insightful, and interesting!
2. To analyze the model alignment during transfer learning from the source domain to similar/distant target domains, the paper uses weight resetting and quantitative channel alignment measurement. These techniques probably inspire others to analyze their GAN models.
3. The paper demonstrates impressive and promising experimental results on image translation and image morphing between different domains. Besides, it demonstrates the benefits of the shared latent space by zero-shot recognition/regression.

##########################################################################

Cons:
1. I’m confused about the example of “Age” in Figure 2. Why does the “Age” change the identity, background, and color largely? It looks like another totally different face.
2. Many details of the results are missing. Take Figure 2 as an example, does the semantic control apply in overlap channels (e.g., 15 for eyebrow in FFHQ2MetFace) or all the distinct channels of the child model (e.g., eyebrow 35)? Are they in the same layer? How do the authors transfer the pose control from face to church and bedrooms in Figure 14? Please check all the results and provide the necessary details.
3. Does the claim of “the hidden latent semantics” hold for the case of (parent FFHQ, child Church, grandchild FFHQ)?
4. I think most of the successful results can be attributed to the well-structured properties of the human face, animal face, and mega face. It would be helpful to provide further discussion on the “real distant domain”, e.g., face VS church. For example, is it helpful to transfer learning from face to church? Are there any other shared semantics between face and church except pose?


**Summary Of The Paper:**


The paper provides interesting analysis and leveraging of GAN’s model alignment (i.e., transfer learning). Without custom architectures and losses, it demonstrates impressive performance in a diverse set of tasks (image translation and image morphing). It also demonstrates promising results for zero-shot image recognition by leveraging the shared latent space of aligned models.


**Summary Of The Review:**


Overall, I vote for accepting. I like the analysis of aligned GANs’ models. The paper also demonstrates impressive experimental results by leveraging the properties of aligned models for image translation, image morphing, and zero-shot classification. My major concern is about the clarity of the paper and some missing details. Hopefully, the authors can address my concern in the rebuttal period.

---

> ### Author Response · Authors · 2021-11-20
> **reply to Reviewer 2pb8**
>
> We thank the reviewer for time and thoughtful feedback. Below we address the main points raised in this review.
>
> * Why does the Age example in Figure 2 change identity, etc.
> The latent direction for Age is obtained through InterfaceGAN [1], which operates in W space. It is well-known that some of the directions discovered by InterfaceGAN are somewhat entangled. The point we’re trying to make in Figure 2 is that the changes induced by different latent directions are similar between the parent (FFHQ) and the children models (Mega and Metface).
>
> * How do we perform the editing in Figure 2 and other figures, like Figure 14?
> For the first 3 attributes in figure 2 (Bangs, Smile, Gaze), we simply manipulate a single channel in S space [2] (the layer and channel numbers are shown under each column). The same layer/channel is manipulated in all three models, and the results appear similar because of alignment. For example, 3_169 (layer 3, channel 169) consistently controls bangs in all three models. For the last 3 attributes (Pose, Age, Gender), we use InterfaceGAN [1] to get the manipulation direction in W space. These directions modify the entire W vectors, which affects all layers and all channels (through affine transformations), but the same manipulation direction is used for each manipulation for all three models. In Figure 14, exactly in the same manner, the latent direction in W that is known to control pose of faces (obtained using InterfaceGAN on FFHQ model), is used *as is* in models fine-tuned to churches (child) and then to bedrooms (grandchild) and still results in a change of pose in these models.
> The channel overlaps reported in Table 1 are unrelated to the manipulation results, they only serve as quantitative evidence of semantic alignment.
>
> * Does the claim of “the hidden latent semantics” hold for the case of (parent FFHQ, child Church, grandchild FFHQ)?
> We believe the answer is yes.
> Due to time limitations, for this rebuttal we could only train the grandchild FFHQ model from the child church model. Extracting all the localized channels reported in Table 1 requires more than a week of computation. Nevertheless, we added a new figure (Figure 34), where we show that the parent and grandchild FFHQ models generate face images with highly similar attributes and identity, given the same input noise vector z.
> In another new figure (Figure 36), we show the same channel still controls the same attribute between parent and grandchild model, even though it had no visible effect in the child Church model.
>
> * Further discussion on the “real distant domain”, e.g., face VS church. For example, is it helpful to transfer learning from face to church? Are there any other shared semantics between face and church except pose?
> Transferring from FFHQ to church increases the model convergence speed, but does not help to improve the FID. As for additional shared semantics, in a new figure (Figure 35), we show that the “beard” direction from FFHQ affects the amount of trees in the church model, and the black hair direction from FFHQ controls the darkness of the building in the church model.
>
>
> 1. Shen, Yujun, et al. "InterFaceGAN: Interpreting the disentangled face representation learned by GANs." IEEE transactions on pattern analysis and machine intelligence (2020).
> 1. Wu, Zongze, Dani Lischinski, and Eli Shechtman. "Stylespace analysis: Disentangled controls for stylegan image generation." Proceedings of the IEEE/CVF Conference on Computer Vision and Pattern Recognition. 2021.

---

### Decision · Program_Chairs · 2022-01-20

**Decision:**

Accept (Oral)

**Comment:**

The paper provides an interesting analysis of aligned GAN models. The paper shows that when a model is obtained (fine-tuned) from another, then the corresponding hidden semantic spaces are aligned. The paper uses this property to show that without any additional architecture or training, the models can perform diverse tasks such as image translation and morphing. The paper also demonstrates that zero-shot tasks can be performed by learning in the parent domain and transferring to the child domain.

All reviewers agree that the paper presents an interesting analysis and findings and will make a valuable contribution to the field. The reviewers raised some particular concerns, which were addressed by the authors in their response.